# VISIR-2: ship weather routing in Python

Gianandrea Mannarini[1], Mario Leonardo Salinas[1], Lorenzo Carelli[1], Nicola Petacco[2], and Josip Orović[3]

[1]Global Coastal Ocean Division, CMCC, via Marco Biagi 5, 73100 Lecce, Italy
[2]DITEN, Università degli Studi di Genova, via Montallegro 1, 16145 Genova, Italy
[3]Maritime Department, University of Zadar, Ul. Mihovila Pavlinovića, 23000 Zadar, Croatia

**Correspondence:** Gianandrea Mannarini (gianandrea.mannarini@cmcc.it)

**Abstract.** Ship weather routing, which involves suggesting low-emission routes, holds potential for contributing to the decarbonisation of maritime transport. However, also for a lack of readily deployable open-source and open-language computational models, its quantitative impact has been explored only to a limited extent.

As a response, the graph-search VISIR model has been refactored in Python, incorporating novel features. For motor vessels, the angle of attack of waves has been considered, while for sailboats the combined effects of wind and sea currents is now accounted for. The velocity composition with currents has been refined, now encompassing leeway as well. Provided that the performance curve is available, no restrictions are imposed on the vessel type. A cartographic projection has been introduced. The graph edges are quickly screened for coast-intersection via a $K$-dimensional tree. A least-$CO_2$ algorithm in the presence of dynamic graph edge weights has been implemented and validated, proving a quasi-linear computational performance. The software suite's modularity has been significantly improved, alongside a thorough validation against various benchmarks. For the visualisation of the dynamic environmental fields along the route, isochrone-bounded sectors have been introduced.

The resulting VISIR-2 model has been employed in numerical experiments within the Mediterranean Sea for the entire 2022, utilising meteo-oceanographic analysis fields. For a 125-meter-long ferry, the distribution of carbon dioxide savings follows a bi-exponential distribution. Routes with a percentage saving of overall $CO_2$ expenditure of at least 2% with respect to the least-distance route were found for prevailing beam or head seas. Two-digit savings, up to 49%, were possible for about ten days in a year. In the case of an 11-meter sailboat, time savings increase with the extent of path elongation, particularly during upwind sailing. The sailboat's routes were made approximately 2.4% faster due to optimisation, with the potential for an additional 0.8% in savings by factoring in currents.

VISIR-2 serves as an integrative model, uniting expertise from meteorology, oceanography, ocean engineering, and computer science, to evaluate the influence of ship routing on the decarbonisation efforts within the shipping industry.

## 1 Introduction

As climate change, with its unambiguous attribution to anthropogenic activities, rapidly unfolds (IPCC, 2023), the causal roles played by various sectors of the economy, as well as the possibilities for mitigation, are becoming more evident. This holds true for the shipping sector as well (IPCC, 2022), which has begun taking steps to reduce its carbon footprint. The International Maritime Organization (IMO) adopted an initial decarbonisation strategy in 2018 (IMO, 2018a) which was later

revised in 2023. The new ambition is to achieve complete decarbonisation by mid-century, addressing all greenhouse gas (GHG) emissions, with a partial uptake of near-zero GHG technology as early as 2030 (IMO, 2023). While no new measures have been adopted yet, the revised strategy is expected to boost efforts to increase the energy efficiency of shipping in the short term (Smith and Shaw, 2023). In line with the European Green Deal, the European Union has adopted new rules to include various GHG (carbon dioxide, methane, and nitrous oxide) emissions from shipping in its Emissions Trading System (EU-ETS), starting from 2024*. For the first time ever, this will entail surrendering allowances for GHG emissions from vessels as well.

Zero-emission bunker fuels are projected to cost significantly more than present-day fossil fuels (Al-Aboosi et al., 2021; Svanberg et al., 2018). Thus, minimising their use will be crucial for financial sustainability. This necessitates energy savings through efficient use, achieved via both technical (e.g., wind-assisted propulsion systems or WAPS) and operational measures (e.g., speed reduction and ship weather routing). According to the "CE-Ship" model, a GHG emissions model for the shipping sector, a global reduction of GHG emissions by 2030 by up to 47% relative to 2008 levels could be feasible through a combination of operational measures, technical innovations, and the use of near-zero-GHG fuels (Faber et al., 2023). A separate study focusing on the European fleet estimates that a reduction of sailing speed alone could potentially lead to a 4-27% emission reduction, while combining technical and operational measures might provide an additional 3-28% reduction (Bullock et al., 2020). The impact of speed optimisation on emissions varies significantly, with potential percentage savings ranging from a median of 20% to as high as 80%, depending on factors such as the actual route, meteorological and marine conditions, and the vessel type (Bouman et al., 2017). In that paper, while cases are reported of up to 50% savings, the role of weather routing was generally assessed to be lower than 10%.

The variability in percentage savings reported in literature can be attributed to the diversity of routes considered, the specific weather conditions, and the type of vessels analysed. Additionally, reviews often use a wide range of bibliographic sources, including grey literature, company technical reports, white papers, and works that fail to address the actual meteorological and marine conditions.

The VISIR (discoVerIng Safe and effIcient Routes, pronunciation: /vi'zi:r/) ship weather routing model was designed to objectively assess the potential impact of oceanographic and meteorological information on the safety and efficiency of navigation. So far, two versions of the model have been released (VISIR-1.a, Mannarini et al. (2016a), and VISIR-1.b, Mannarini and Carelli (2019a)). However, the use of a proprietary coding language (Matlab) may hinder its further adoption. Also, the experience with VISIR-1 suggested the need to enhance the modularity of the package and implement other best practices of scientific coding, as recommended in Wilson et al. (2014). Another area where innovation seemed possible was the development of a comprehensive framework to perform weather routing for both motor vessels and sailboats. While some aspects of this requirement were covered through a more modular approach, it also involved rethinking how to utilise environmental fields such as waves, currents, and wind. Furthermore, while the carbon intensity savings of least-time routes were already estimated via VISIR-1.b, a dedicated algorithm for the direct computation of least-$CO_2$ routes was lacking.

---

*https://climate.ec.europa.eu/eu-action/transport-emissions/reducing-emissions-shipping-sector_en (Interinstitutional negotiations)

To address all these requirements, we designed, coded, tested, and conducted extensive numerical experiments with the VISIR-2 model (Salinas et al., 2024a). VISIR-2 is a Python-coded software, inheriting from VISIR-1 the fact that it is based on a graph-search method. However, VISIR-2 is a completely new model, leveraging the previous experience, while also offering many new solutions and capabilities. Part of the validation process made use of its ancestor model VISIR-1. The computational performance has been enhanced, and efforts have been made to improve usability. In view of use for routing of WAPS vessels, we have developed a unified modelling framework, accounting for involuntary speed loss in waves, advection via ocean currents, thrust and leeway from wind. A novel algorithm for finding the least-$CO_2$ routes has been devised, building upon Dijkstra's original method. The visualisation of dynamic information along the route has been further improved by using isochrone-bounded sectors. VISIR-2 features are thoroughly described in this paper, along with some case studies and hints for possible development lines in the future.

The VISIR-2 model enabled systematic assessment of $CO_2$ savings deriving from optimal use of meteo-oceanographic information in the voyage planing process. Different from most existing works cited above, the emissions could clearly be related to significant wave height, relative angle of waves, and also engine load factor. This is discussed further in Sect. 6.1. Ocean currents can be added to the optimisation, providing more realistic figures for the emission savings.

VISIR-2 is a numerical model well-suited for both academic and technical communities, facilitating the exploration and quantification of ship routing's potential for operational decarbonisation. In the European Union, there may be a need for independent verification of emissions reported in the monitoring, reporting, and verifying (MRV) system[†]. Utilising baseline emissions computed through a weather routing model could enhance the accuracy and reliability of this verification process. Additionally, the incorporation of shipping into the EU-ETS intensifies the importance of minimising GHG emissions. Internationally, even with the adoption of zero- or low-carbon fuels encouraged by the recent update of the IMO's strategy, it remains critical to save these fuels as much as possible. Therefore, VISIR-2 could potentially serve as a valuable tool for saving both fuel and allowances.

The remainder of this paper comprises a literature investigation in Sect. 1.1, followed by an in-depth presentation of the innovations introduced by VISIR-2 in Sect. 2. Subsequently, the model validation is discussed in Sect. 3 and its performance is assessed in Sect. 4. Several case studies in the Mediterranean Sea follow (Sect. 5). The results on the $CO_2$ emission percentage savings, some potential use cases, and an outlook of possible developments of VISIR-2 are discussed in Sect. 6. Finally, the conclusions are presented in Sect. 7. An Appendix contains technical information regarding the computation of the angle of attack between ship's heading and course (App. A), as well as details about the neural network employed to identify the vessel performance curves (App. B).

## 1.1 Literature review on weather routing

This compact review of systems for ship weather routing will be limited to web applications (Sect. 1.1.1) and peer-reviewed research papers (Sect. 1.1.2). It is further restricted to the free available versions of desktop applications, while the selection

---
[†]https://www.emsa.europa.eu/thetis-mrv.html

of papers is meant to update the wider reviews already provided in Mannarini et al. (2016a); Mannarini and Carelli (2019b). A critical gap analysis (Sect. 1.1.3) completes this subsection.

### 1.1.1 Web applications

FastSeas[‡] is a weather routing tool for sailboat, with editable polars, and possibility to consider a motor propulsion. Wind forecasts are taken from the National Oceanic and Atmospheric Administration Global Forecast System (NOAA GFS) model and ocean surface currents from NOAA Ocean Surface Current Analyses Real-time (OSCAR). It makes use of Windy.com imagery, a free choice of endpoints is offered, departure time can vary between present and a few days in the future, the voyage-plan is exportable in various formats.

The AVALON web router[§] provides a coastal weather routing service for sailboats, within subregions of France, United Kingdom, and United States. It offers a choice among tens of sailboats, with the option to consider also a motor-assisted propulsion. Hourly departure dates within a couple of days are allowed, and ocean weather along the routes is provided in tabular form.

GUTTA-VISIR[¶] is a weather routing tool for a specific ferry developed in the frame of the "savinG fUel and emissions from mariTime Transport in the Adriatic region" (GUTTA) project. It provides both least-time and least-$CO_2$ routes between several ports of call. It makes use of operational wave and current forecast fields from the Copernicus Marine Environment Monitoring Service (CMEMS). The route departure date and time and the engine load can be varied. Waves, currents, or isolines can be rendered along with the routes, which can be exported.

openCPN[‖] is a comprehensive open-source platform, including also a weather routing tool for sailboats. The product name originates from "open-source Chart Plotter Navigation". The vessel performance curves are represented via polars, and forecast data in grib format or a climatology can be used. Nautical charts can be downloaded and integrated into the graphical user interface. The programming language is C++ and a velocity composition with currents is accounted for. A detailed documentation of the numerical methods used is lacking, though.

### 1.1.2 Research papers

A review of ship weather routing methods and applications was provided by Zis et al. (2020). Several routing methods such as the isochrone method, dynamic programming, calculus of variations, and pathfinding algorithms were summarised, before a taxonomy of related literature was proposed. The authors made the point that the wide range of emission savings reported in literature might in future be constrained via defining a baseline case, providing benchmark instances, and performing sensitivity analyses, e.g. on the resolution of the environmental data used.

A specific weather routing model was documented by Vettor and Guedes Soares (2016). It integrates advanced seakeeping and propulsion modelling with multi-objective (fuel consumption, duration, and safety) optimisation. An evolutionary algo-

---

[‡]https://fastseas.com/
[§]https://www.webrouter.avalon-routing.com/compute-route
[¶]https://gutta-visir.eu
[‖]https://opencpn.org/

rithm was used, with initialisation from both random solutions and single-objective routes from a modified Dijkstra's algorithm. Safety constraints were considered via probabilities of exceeding thresholds for slamming, green water, or vertical acceleration. A specific strategy was proposed to rank solutions within a Pareto frontier.

A stochastic routing problem was addressed in Tagliaferri et al. (2014). A single upwind leg of a yacht race is considered, with the wind direction being the stochastic variable. The vessel was represented in terms of polars and the optimal route was computed via dynamic programming. The skipper's risk propensity was modelled via a specific preference on the wind transition matrices. This way it was shown how the risk attitude affects the chance to win a race.

Ladany and Levi (2017) developed a dynamic programming approach to sailboat routing which accounts for the tacking time. The latter was assumed to be proportional to the amplitude of the course change. Furthermore, a sensitivity analysis was conducted, considering both the uncertainty on wind direction and the magnitude of the discretisation step in the numerical solution.

Sidoti et al. (2023) provided a consistent framework for a dynamic programming approach for sailboats, considering both leeway and currents. In order to constrain the course of the boat on the available edges on the numerical grid, an iterative scheme was adopted. Case studies with NAVGEM winds and Global HYCOM currents were carried out in a region across the Gulf Stream. The results without leeway were validated versus openCPN.

The impact of stochastic uncertainty on WAPS ships was addressed by Mason et al. (2023b). A dynamic programming technique was used, and "a priori" routing (whereby information available at the start of the journey only is used) was distinguished from "adaptive" routing (whereby the optimum solution is updated based on information that becomes available every 24 h along a journey). The latter strategy is shown, for voyages lasting several days, to be more robust with respect to the unavoidable stochastic uncertainty of the forecasts.

### 1.1.3 Knowledge gap

A few open web applications exist, mainly for sailboats, and with limited insight into the numerical methods. Case studies results from weather routing systems developed in the academia were published, but (exception of Mason et al. (2023b)) no systematic assessment of $CO_2$ savings was provided. Furthermore, no related software was disclosed in any case. The openCPN package lacks both a peer-reviewed publication and a documentation of the methods implemented. A prevalence of dynamic programming approaches is noted, especially for web applications, with graph-search method being used in research papers only. The tools either focus on sailboats (with or without a motor) or on motor vessels. When both are available (such as in Fastseas), the motor vessel is described in terms of polars.

From this assessment, an open-source and well-documented ship weather routing model, for both motor vessels and sailboats, with flexible characterisation of vessel performance, appears as a gap which the present work aims to close.

## 2  Technical advancements

This section includes a revision of the vessel kinematics of VISIR, as given in Sect. 2.1; changes in the graph generation procedure and in the use of static environmental fields in Sect. 2.2; updates to the computation of graph edge weights in Sect. 2.3; an additional optimisation objective in the shortest path algorithm in Sect. 2.4; new vessel performance models in Sect. 2.5; innovative visualisation capabilities in Sect. 2.6; and a more modular structure of the software package, presented in Sect. 2.7. Further technical details of the VISIR-2 code are presented in the software manual, provided along with its online repository (Salinas et al., 2024a).

### 2.1  Kinematics

For a graph-search model such as VISIR to deal with waves, currents, and wind, for various vessel types, several updates to its approach for velocity composition were needed. They included both generalisations and use of new quantities, addressed in this subsection, as well as a new numerical solution, addressed in App. A.

As in Mannarini and Carelli (2019b), the kinematics of VISIR-2 is based on both the principle of superposition of velocities and the discretized sailing directions existing on a graph. However, while previously the vessel's speed over ground (SOG) was obtained from the magnitude of the vector sum of speed through water ($\overrightarrow{T} = \text{STW } \hat{t}$) and ocean current ($\overrightarrow{w}$), we here show that, more generally, SOG is the magnitude of a vector $\overrightarrow{G} = \text{SOG}\,\hat{e}$ being the sum of the forward speed $\overrightarrow{F} = \text{F } \hat{h}$ and an effective current $\overrightarrow{\omega}$, and both will be defined in the following. In the absence of leeway, the $\overrightarrow{F}$ and $\overrightarrow{T}$ vectors are identical and $\overrightarrow{\omega} = \overrightarrow{w}$, so the newer approach encompasses the previous one.

Making reference to Fig. 1, the use of a graph constrains the vessel's course over ground to be along $\hat{e}$, being the orientation of one of the graph's arcs. Thus, any cross-component of velocity, or along a $\hat{o}$ versor such that $\hat{e} \cdot \hat{o} = 0$, must be null. This implies that, to balance the cross flow from the currents, the vessel must head into a direction $\hat{h}$ slightly different from the course $\hat{e}$. In Mannarini and Carelli (2019b) such an angle of attack was defined as

$$\delta = \psi_h - \psi_e \tag{1}$$

where for both $\psi_h$ (heading, or HDG), $\psi_e$ (course over ground, or COG) a nautical convention is used (due North, to-direction). That framework is here generalised to also deal with the vector nature of some environmental fields, such as waves or wind. Using a meteorological convention (due North, from-direction) for both the $\psi_a$ (waves) and $\psi_i$ (wind) directions, we here introduce the $\delta_f$ angles, defined as

$$\delta_f = \psi_h - \psi_f = \delta - \gamma_f \tag{2}$$

where $\gamma_f = \psi_f - \psi_e$ are the relative angles between the $f$ environmental field and the ship's course. $f = a$ for waves and $f = i$ for wind. In computing angular differences, their convention should be considered (see `angular_utils.py` function in the VISIR-2 code). Thus, $\delta_f = 0$ whenever the ship heads into the direction from which the field comes, and $\gamma_f = 0$ if her course is into such a direction.

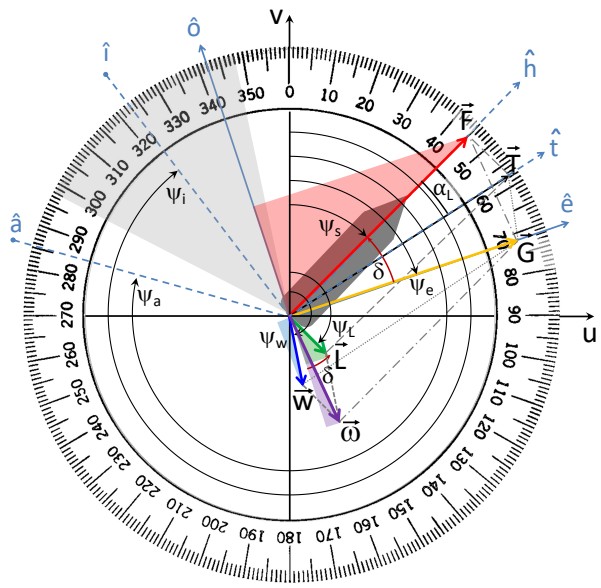

**Figure 1.** Angular configuration with ($\delta = -27°$, $\gamma_i = -109°$), resulting in $\epsilon = +1$. The dark grey area represents the ship hull, while the light grey shaded area denotes the no-go zone for $\alpha_0 = 25°$. Clockwise-oriented arcs indicate positive angles, and filled circles at line ends denote the meteorological ("from") convention.

Furthermore, we here define leeway as a motion, caused by the wind, transversal to the ship's heading. From the geometry shown in Fig. 1, the oriented direction of leeway $\psi_L$ is given by

$$\psi_L = \psi_h + \epsilon \cdot 90° \tag{3}$$

with

$$\epsilon = \cos\left(180° \lfloor \delta_i/180° \rfloor\right) \tag{4}$$

where the $\lfloor \cdot \rfloor$ delimiters indicate the floor function. Thus, $\epsilon$ is positive for $\delta_i$ in the $[0, 180°]$ range and flips every $180°$ of the argument. This is not the sole possible definition of leeway. For instance, in Breivik and Allen (2008) a distinction between downwind and crosswind component of leeway is made. However, the present definition is consistent with the subsequent data 190   of vessel performance (Sect. 5.2.2).

Upon expressing the magnitudes $F$ of vessel's forward velocity and $L$ of leeway velocity as

$$F = F(|\delta_i|, V_i; |\delta_a|, H_s, \chi) \tag{5a}$$
$$L = L(|\delta_i|, V_i), \tag{5b}$$

with wind magnitude $V_i$, significant wave height $H_s$, and engine load $\chi$, a leeway angle, being the ship's heading change between the $\overrightarrow{F}$ and $\overrightarrow{T}$ vectors, can be defined as

$$\alpha_L = \mathrm{atan}(L/F) \tag{6}$$

The above introduced $\alpha_L$ is not a constant but depends on both wind magnitude $V_i$ and the module of the relative direction, $|\delta_i|$. Also, from Fig. 1, it is seen that $F = \mathrm{STW} \cdot \cos\alpha_L$. Thus, in the absence of leeway ($\alpha_L = 0$), one retrieves the identity of $F$ and STW which was an implicit assumption done in Mannarini and Carelli (2019b).

Eq. 5 is a major innovation with respect to the formalism of Mannarini and Carelli (2019b), as an angular dependence in the performance curve is introduced in VISIR also for motor vessels for the first time (Sect. 2.5). Eq. 5a include dependencies on both wind and waves. Furthermore, a possible dependency on $\chi$, the fractional engine load (or any other propulsion parameter), is here highlighted.

Within this formalism, if the vessel is a sailboat (or rather a motor vessel making use of WAPS), just an additional condition should be considered. That is, given the wind-magnitude dependent no-go angle $\alpha_0(V_i)$, upwind navigation is not permitted, or:

$$|\psi_h - \psi_i| \notin [-\alpha_0, \alpha_0] \tag{7}$$

Now, given a water flow expressed by the vector:

$$\overrightarrow{w} = C\,\hat{w} = (u, v)^T, \tag{8}$$

and making reference to Fig. 1, the flow projections along ($\hat{e}$) and across ($\hat{o}$) the vehicle course respectively are

$$w_\parallel = C\cos(\psi_e - \psi_w) = u\sin(\psi_e) + v\cos(\psi_e), \tag{9a}$$
$$w_\perp = C\sin(\psi_e - \psi_w) = v\sin(\psi_e) - u\cos(\psi_e), \tag{9b}$$

where also for the ocean flow direction $\psi_w$ the nautical convention is used.

In analogy to Eq. 9, and using nautical convention also for $\psi_L$, the along and cross-course projection of the leeway are given by

$$w_\parallel^{(L)} = L\cos(\psi_e - \psi_L) = -\epsilon\, L\sin\delta, \tag{10a}$$
$$w_\perp^{(L)} = L\sin(\psi_e - \psi_L) = -\epsilon\, L\cos\delta. \tag{10b}$$

The simple relations on the r.h.s. of Eq. 10 follow from the similitude of the red and green-shaded triangles in Fig. 1. As $\delta$ typically is a small angle (cf. App. A), it is apparent that the cross component of the leeway, $w_\perp^{(L)}$, is the dominant one. Its sign is such that it is always downwind, see Fig. 1. If relevant, the Stokes' drift (van den Bremer and Breivik, 2018) could be treated akin to an ocean current, and one would obtain for its projections a couple of equations formally identical to Eq. 9.

Finally, the components of the effective flow $\boldsymbol{\omega}$ advecting the vessel are

$$\omega_\parallel = w_\parallel + w_\parallel^{(L)}, \tag{11a}$$

$$\omega_\perp = w_\perp + w_\perp^{(L)}. \tag{11b}$$

Due to Eq. 10, both $\omega_\parallel$ and $\omega_\perp$ are functions of $\delta$. We recall that the "cross" and "along" specifications refer to vessel course $\psi_e$, differing from vessel heading by the $\delta$ angle.

   The graphical construction in Fig. 1 makes it clear that $\overrightarrow{G}$ = SOG $\hat{e}$ equals $\overrightarrow{T} + \overrightarrow{w}$ or $\overrightarrow{F} + \overrightarrow{\omega}$. Using the latter equality, together with the course assignment condition, and projecting along both $\hat{e}$ and $\hat{o}$, two scalar equations are obtained, namely:

$$SOG = F\cos(\delta) + \omega_\parallel, \tag{12a}$$

$$0 = -F\sin(\delta) + \omega_\perp, \tag{12b}$$

Eq. 12 are formally identical to those found in Mannarini and Carelli (2019b) in presence of ocean currents only. This fact suggests the interpretation of $\omega$ as an effective current.

   However, Eq. 12 alone is no more sufficient to determine the ocean current vector $\overrightarrow{w}$. In fact, it is mingled with the effect of wind through leeway, to form the effective flow $\overrightarrow{\omega}$ (Eq. 11). This is why, in presence of strong winds, reconstruction of ocean

currents from data of COG and HDG via a naive inversion of Eq. 12 is challenging. This was indeed found by Le Goff et al. (2021) using Automatic Identification System (AIS) data across the Agulhas Current.

   As it reduces the ship's speed available along its course (Eq. 12a), the angle of attack $\delta$ plays a pivotal role in determining the SOG. However, in presence of an angle-dependent vessel speed (Eq. 5a), $\delta$ is no more given by a simple algebraic equation corresponding to Eq. 12b as in Mannarini and Carelli (2019b), but by a transcendental one:

$$\sin\delta = \frac{\omega_\perp(\delta, \delta_i(\delta))}{F(|\delta_i(\delta)|, |\delta_a(\delta)|)} \quad \Leftrightarrow \quad F \neq 0. \tag{13}$$

In fact, due to Eq. 2, the r.h.s. of Eq. 13 depends both explicitly and implicitly on $\delta$. Just in the limiting case of null currents, Eq. 6, Eq. 10b and Eq. 12b collectively imply that

$$\delta = -\epsilon\,\alpha_L \tag{14}$$

However, in general, the actual value of the forward speed $F$ is only determined once the $\delta$ angle is retrieved from Eq. 13.

To our knowledge, the transcendental nature of the equation defining $\delta$ had not been pointed out previously. Furthermore, in VISIR-2 an efficient numerical approximation for its solution is provided, see App. A. This is also a novelty with practical benefits for the development of efficient operational services based on VISIR-2.

   We note that Eq. 13 holds if and only if

$$|\omega_\perp| \leq F. \tag{15}$$

Should this not be the case, the vessels's forward speed would not balance the effective drift.

As $F$ is always non-negative, Eq. (13) implies that $\text{sgn}(\delta) = \text{sgn}(\omega_\perp)$. In particular, in the case of an effective crossflow $\omega_\perp$ bearing, as in Fig. 1, to starboard, a counterclockwise change of vehicle heading ($\delta < 0$) is needed for keeping course (note $\hat{o}$ versor's orientation in Fig. 1).

Eq. 12 can be solved for the speed over ground SOG, which reads

$$SOG = \omega_\parallel + \sqrt{F^2 - \omega_\perp^2}. \tag{16}$$

According to Eq. 16 the crossflow $\omega_\perp$ always reduces the SOG, as part of vehicle momentum must be spent for balancing the drift. The along-edge flow $\omega_\parallel$ (or "effective drag") may instead either increase or decrease SOG.

Finally, given that $\overrightarrow{G} = \mathrm{d}\boldsymbol{x}/\mathrm{d}t$, by taking the module of the left side, and approximating the right-hand side (r.h.s.) with its finite-difference quotient, the graph edge weight $\delta t$ is computed as

$$\delta t = \frac{\delta x}{SOG}, \tag{17}$$

where $\delta x$ is the edge length and SOG is given by Eq. 16. As the environmental fields determining the SOG are both space and time dependent, the weights $\delta t$ are computed via the specific interpolation procedures in Sect. 2.3, and the shortest paths via the algorithms provided in Sect. 2.4.

From Eq. 17, it follows that the condition

$$SOG \geq 0 \tag{18}$$

should be checked in case the specific graph-search method used does not allow for use of negative edge weights (as is the case for the Dijkstra's algorithm, cf. Bertsekas (1998)). Violation of Eq. 18 may occur in presence of a strong counter-flow $\omega_\parallel$ along a specific graph edge, which weight must correspondingly be set to not-a-number.

The $CO_2$ emissions along a graph edge are given by

$$\delta CO_2 = \Gamma \, \delta t, \tag{19}$$

where $\Gamma = \Gamma(|\delta_i|, V_i; |\delta_a|, H_s, \chi)$ is the $CO_2$ emission rate of the specific vessel in presence of the actual meteo-marine conditions. Both $\delta t$ and $\delta CO_2$ are used in the least-$CO_2$ algorithm introduced in Sect. 2.4.

## 2.2 Graph generation

Graph preparation is crucial for any graph-search method. Indeed, graph edges represent potential route legs. Therefore, the specific edges included within the graph directly influence the route topology. In addition, as it will be shown in Sect. 2.3.2, the length of edges affects the environmental field value represented by each edge. On the other hand, graph nodes determine the accessible locations within the domain. Therefore, they must be selected with consideration for both the presence of landmasses and shallow waters.

The structure of the mesh is also the most fundamental difference between a graph-search method (such as Dijkstra's or $A^*$) and dynamic programming. Indeed, a dynamic programming problem can be transformed into a shortest path problem on a

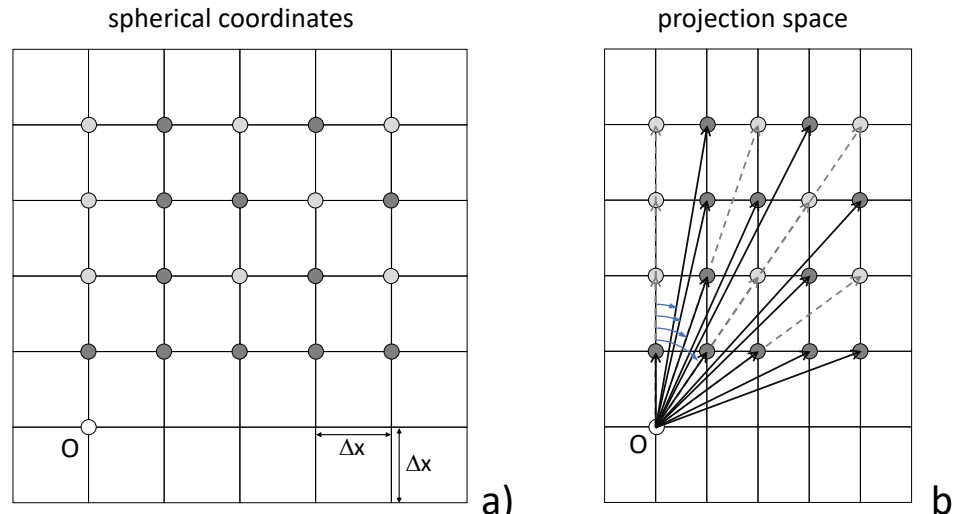

**Figure 2.** Graph stencil for $\nu = 4$: a) grid in spherical coordinates with $\Delta x$ resolution along both latitude and longitude axes; b) Mercator projection, with different resolution along $y$ or $x$, and graph edges (black thick and grey dashed lines) and angles (light blue) relative to due North. The $y$ spacing is here shown as constant but, over a large latitudinal range, does vary. In VISIR-2, just the $N_{q1}(\nu)$ dark grey nodes (cf. Eq. 20 and Tab. 1) are connected to the origin while, in VISIR-1, all $\nu(\nu + 1)$ dark or light grey nodes were connected.

graph (Bertsekas, 1998)[Sect.2.1]. In the former, the nodes are organised along sections of variable amplitude corresponding to stages. In the latter, nodes can uniformly cover the entire domain. However, for both dynamic programming (Mason et al., 2023b) and graph-methods (Mannarini et al., 2019b), the number of edges significantly impacts the computational cost of computing a shortest path.

To efficiently address all these aspects, VISIR-2's graph preparation incorporates several updates compared to its predecessor, VISIR-1b, as outlined in the following sub-sections.

### 2.2.1 Cartographic projection

The nautical navigation purpose of a ship routing model necessitates the provision of both length and course information for each route leg. On the other hand, environmental fields used to compute ship's SOG (Eq. 16) are typically provided as a matrix
of values in spherical coordinates (latitude and longitude, similar to the fields used in Sect. 5.1). To address these aspects, VISIR-2 introduces a projection feature, which was overlooked in both VISIR-1.a and VISIR-1.b. The Mercator projection was chosen for its capacity to depict constant bearing lines as straight lines (Feeman, 2002). The implementation was carried out using the `pyproj` library, which was employed to convert spherical coordinates on the WGS-84 (World Geodetic System 1984) ellipsoid into a Mercator projection, with the equator serving as the standard parallel. For visualisation purposes, both
the `matplotlib` and `cartopy` libraries were utilised.

### 2.2.2 Edge direction

Leg courses of a ship route originate from graph edge directions. To determine them, we consider the Cartesian components of the edges in a projected space. As seen from Fig. 2, this approach results in smaller angles relative to due North compared to using an unprojected graph. For the graph ($\nu$, $\Delta x$) values and average latitude of the case studies considered in this paper (Sect. 5), the maximum error between an unprojected graph and a Mercator projection would be about $5°$(cf. Tab. 1). However, this angle critically impacts vessels like sailboats, whose performance hinges on environmental field orientation. Additionally, at higher latitudes, the courses tend to cluster along meridional directions. To achieve a more isotropic representation of courses, constructing the graph as a regular mesh in the projected space would be needed, which is left for future refinements of VISIR-2.

For a given sea domain, a graph is typically computed once, and subsequently utilised for numerous different routes. However, in VISIR-1 edge direction was recalculated every time a graph was utilised. In VISIR-2 edge direction is computed just once, during graph generation, after which it is added to a companion file of the graph (`edge_orientation`). In the definition of the edge direction, the nautical convention (due North, to-direction) is used in VISIR-2 graphs.

Finally it should be noted that in a directed graph, such as VISIR-2's, also edge orientation matters. Each edge acts as an arrow, conveying flow from "tail" to "head" nodes. Edge orientation refers to the assignment of the head or tail of an edge. Orientation holds a key to understanding vessel performance curves, explored further in Sect. 2.5.

### 2.2.3 Quasi-collinear edges

A further innovation regarding the graph involves an edge pruning procedure. This eliminates redundant edges that carry very similar information. Indeed, some edges have the same ratio of horizontal to vertical grid hops (see `possible_hops()` function). Examples of these edges are the solid and dashed arrows in Fig. 2.b. When the grid step size and the number of hops are not too large, these edges point to nearly the same angle relative to North. However, starting from an equidistant grid in spherical coordinates, the vertical spacing in Mercator projection is uneven. Thus, the directions of those edges are not exactly the same, and we call such edges "quasi-collinear". In VISIR-2, only the shortest one among these quasi-collinear edges is retained. This corresponds to the solid arrows in Fig. 2.b. This reduces the number $N_{q1}$ of edges within a single quadrant to

$$N_{q1}(\nu) = 2\sum_{k=1}^{\nu} \varphi(k) \leq \nu(\nu+1) \tag{20}$$

where $\varphi$ is Euler's Totient function and $\nu$ the maximum number of hops from a given node of the graph. Thus, the quantity right of the inequality represents the total number of edges of a quadrant, including quasi-collinear ones. Using Eq. 20, already at $\nu = 4$ more than one third of all edges get pruned, and, at $\nu = 10$, nearly half of them (cf. Tab. 1).

This benefits both the computer memory allocation and the computing time for the shortest path. The latter is linear in the number of edges, cf. Bertsekas (1998)[Sect.2.4.5]. A further benefit of pruning quasi-collinear edges is a more faithful representation of the environmental conditions. In fact, the environmental field's values at the graph nodes are used for estimating the edge weights (see Sect. 2.3). Thus, keeping just shorter edges avoids using a less spatially resolved information.

### 2.2.4 Bathymetry and draught

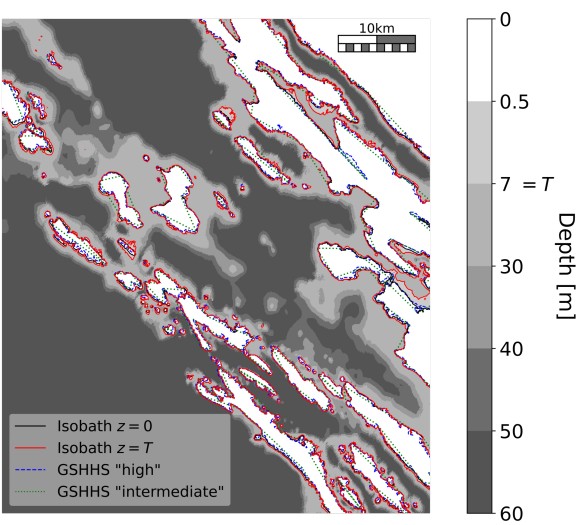

**Figure 3.** Bathymetry field from EMODnet represented in shades of grey, with contour lines at depths at $z = 0$ m and $z = T$, where $T = 7$ m is the vessel draught. Additionally, the GSHHS shoreline at two different spatial resolutions is included.

The minimal safety requirement is that navigation does not occur in shallow waters. It corresponds to the condition that vessel draught $T$ does not exceed sea depth $z$ at all graph edges used for the route computations. This is equivalent to a positive under keel clearance $UKC = z - T$. As explained later in Sect. 2.3.2, this can be checked by either evaluating the average $UKC$ at the two edge nodes, or by interpolating it at the edge barycentre.

However, for some specific edge, $UKC$ could still be positive and the edge cross the shoreline. This is avoided in VISIR by checking for mutual edge – shoreline crossings. Given the burden of this process, in VISIR-1b a procedure for restricting the check to inshore edges was introduced. In VISIR-2, as envisioned in Mannarini and Carelli (2019b)[App.C], the process of searching for intersections is carried out using a $K$-dimensional Tree (KDT, Bentley (1975); Maneewongvatana and Mount (1999)). This is a means of indexing the graph edges via a spatial data structure which can effectively be queried for both nearest neighbours (coast proximity of nodes) and range queries (coast intersection of edges). The `scipy.spatial.KDTree` implementation was used[**]. Use of an advanced data structure also in the graph is a novelty of VISIR-2.

Various bathymetric databases can be used by VISIR-2. For European seas, the EMODnet dataset[††] (1/16 arc-minute resolution or about 116 m) was used while, for a global coverage, the GEBCO_2022[‡‡] (15 arc-second resolution or about 463 m) is available.

---

[**]https://docs.scipy.org/doc/scipy/reference/

[††]https://portal.emodnet-bathymetry.eu/

[‡‡]https://www.gebco.net/data_and_products/gridded_bathymetry_data/

### 2.2.5 Shoreline

The bathymetry dataset, if detailed enough, can even be used for deriving an approximation of the shoreline. From Fig. 3 it
is seen that a "pseudo-shoreline" derived from the UKC=0 contour line of a fine-enough bathymetry (the EMODnet one) can
effectively approximate an official shoreline (the GSHHG[§§] one, at the "high"-type resolution of 200 m).

Such pseudo-shoreline is the one used in VISIR-2 for checking the edge crossing condition specified in Sect. 2.2.4.

### 2.3 Edge weights

For computing shortest paths on a graph, its edge weights are preliminarily needed. Due to Eq. 16 and Eq.5a-5b, they depend
on both space- and time-dependent environmental fields, which information has to be remapped to the numerical grids of
VISIR-2. This is done in a partly differently way than in VISIR-1, providing users with improved flexibility and control over
numerical fields. These novel options are documented in what follows.

#### 2.3.1 Temporal interpolation

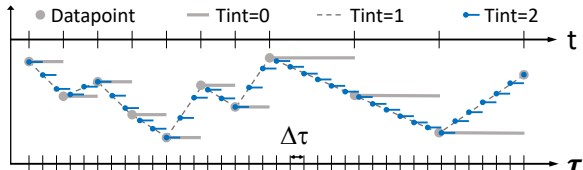

**Figure 4.** Temporal grid of VISIR-2. The upper horizontal axis ($t$) represents the coarse and uneven time resolution of the original environmental field. The lower horizontal axis corresponds to the fine and even time grid with resolution $\Delta\tau$ to which it is remapped.

The time at which edge weight are evaluated is key to the outcome of the routing algorithm. In Mannarini et al. (2019b), to
355 improve on coarse time resolution of the environmental field, a linear interpolation in time of the edge weight was introduced
("Tint = 1" option in Fig. 4). In VISIR-2 instead the environmental field values (grey dots) are preliminarily interpolated in time
on a finer grid with $\Delta\tau$ spacing ("Tint = 2" or blue dots). Then, the edge weight at the nearest available timestep (`np.floor`
function used, corresponding to the blue segments) is selected.

#### 2.3.2 Spatial interpolation

The numerical environmental field and the graph grid may possess varying resolutions and projections. Even if they were
identical, the grid nodes might still be staggered. Moreover, it is necessary to establish a method for assigning the field values
to the graph edges. For all these reasons, spatial interpolation of the environmental fields is essential.

---

[§§]https://www.ngdc.noaa.gov/mgg/shorelines/

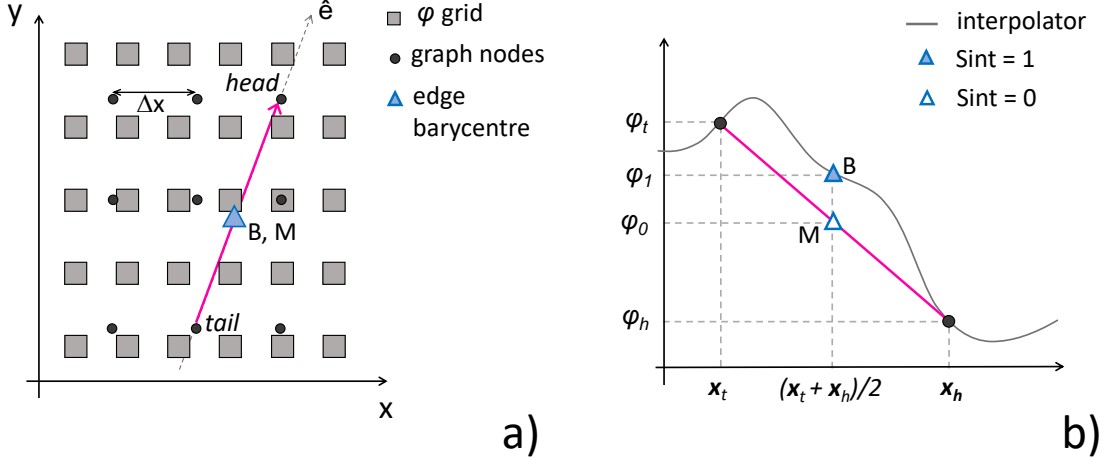

**Figure 5.** Spatial interpolation in VISIR-2. a) The squares represent grid nodes of the environmental field $\varphi(\boldsymbol{x})$, and the filled circles graph grid nodes. A graph edge is depicted as a magenta segment. b) Transect of a) along the edge direction $\hat{e}$, with the interpolator of $\varphi$ as a grey solid line. The (0,1) subscripts refer to the value of the Sint parameter while $h$ and $t$ to the edge head and tail, respectively.

VISIR-2 first computes an interpolant using the `scipy.interpolate.interp2d` method. Then, to assign a representative value of the $\varphi$ field on an edge, two options are available, which are depicted in Fig. 5. In the first one or "Sint = 0",

an average between the edge head and tail's values is computed, $\varphi_0 = (\varphi_h + \varphi_t)/2$. In the second one or "Sint = 1", the field interpolator is evaluated at the location of the edge barycentre, $\varphi_1 = \varphi\left((\boldsymbol{x}_h + \boldsymbol{x}_t)/2\right)$. As the field generally is nonlinear, this leads to different outcomes.

The two interpolation schemes were tested with various non-convex functions and some results are reported in Sect.S0 of the Supplement . We observed that both options converge to a common value as the $(1/\Delta x)$ resolution of the graph grid increases.

Thus, both options benefit from pruning collinear edges (Sect. 2.2.3), as they remove some longer edges from the graph, but neither demonstrates consistent superiority over the other in terms of fidelity.

However, setting Sint = 0 results in higher computational efficiency. This is because the interpolator is applied at each node with Sint = 0, whereas it is applied at each edge with Sint = 1. Given that the number of edges exceeds the number of nodes by a factor defined by Eq. 20, for the case studies ($4 \leq \nu \leq 5$), the computational time for Sint = 0 was approximately one order

of magnitude smaller than with Sint = 1. Therefore, the latter was chosen as the default for VISIR-2.

The resulting edge-representative environmental field value affects the edge delay (Eq. 17), consequently impacting the vessel's speed as per Eq. 5. Hence, a nonlinear relationship exists between the field value and the local sailing speed. Thus, the actual choice of the Sint parameter is not expected to systematically bias the vessel's speed in either direction.

Even before the spatial interpolation is performed, the so called "sea-over-land" extrapolation is applied to the marine fields.

This step, which is needed for filling the gaps in the vicinity of the shoreline, is conceptually performed as in Mannarini et al. (2016a)[Fig.7] and implemented in the `seaoverland` function.

Since wave direction is a circular-periodic quantity, we calculate its average using the circular mean[¶] ahead of interpolation. This differs from wind direction, typically given as Cartesian components $(u,v)$, which can be interpolated directly.

## 2.4 Shortest path algorithms

A major improvement made possible by the Python coding of VISIR-2 is the availability of built-in, advanced data structures such as dictionaries, queues, and heaps. They are key in the efficient implementations of graph-search algorithms (Bertsekas, 1998). In particular, as data structures are used, Dijkstra's algorithm worst case performance can improve from quadratic, $\mathcal{O}(N^2)$, to linear-logarithmic, $\mathcal{O}(N \log N)$, where $N$ is the number of graph nodes.

Nonetheless, Dijkstra's original algorithm exclusively accounted for static edge weights (Dijkstra, 1959). When dynamic
edge weights are present, (Orda and Rom, 1990) demonstrated that, in general, there are no computationally efficient algorithms. However, they also showed that, upon incorporating a waiting time at the source node, it is possible to keep the algorithmic complexity of a static problem. Given the assumption that the rate of variation of the edge delay $\delta t$ satisfies

$$\frac{d}{dt}\delta t \geq -1 \tag{21}$$

an initial waiting time is not even unnecessary. This condition was assumed to hold in Mannarini et al. (2016a) for implement-
395 ing a time-dependent Dijkstra's algorithm. That version of the shortest path algorithm could not be used with an optimisation objective differing from voyage duration. As one aims to compute e.g. least-$CO_2$ routes, the algorithm requires further generalisation. This has been addressed in VISIR-2 via the pseudocode provided in both Alg. 1 and Alg. 2. For its implementation in Python, we made use of a modified version of the `single_source_Dijkstra` function of the `networkX` Python library. The modification consisted in retrieving an edge weight at a specific time step. This is achieved via Alg. 2. Thereto, the
400 $cost.at\_time$ pseudo-function represents a `networkX` method to access the edge weight information.

We note the generality of the pseudocode with respect to the edge weight type ($wT$ parameter in both Alg. 1 and Alg. 2). This implies that the same algorithm could also be used to compute routes minimising figures of merit differing from $CO_2$ , such as different GHG emissions, cumulated passenger comfort indexes, or total amount of underwater radiated noise. This hinges solely on the availability of sufficient physical-chemical information to compute the corresponding edge weight in relation to
405 the environmental conditions experienced by the vessel. A flexible optimisation algorithm is yet another novelty of VISIR-2.

The shortest-distance and the least-time algorithm invoked for both motor vessels and sailboats are identical. Differences occur at post-processing level only, as different dynamical quantities (related to the marine conditions or the vessel kinematics) have to be evaluated along the optimal paths. Corresponding performance differences are evaluated in Sect. S1 of the Supplement .

## 410 2.4.1 non-FIFO

As previously mentioned, the Dijkstra's algorithm can recover the optimal path in the presence of dynamic edge weights, if Eq. 21 is satisfied. In cases where the condition is not met, Orda and Rom (1990) presented a constructive method for

---

[¶]https://en.wikipedia.org/wiki/Circular_mean

## Algorithm 1 _DIJKSTRA_TDEP

**Input:** $(G, source, target, wT, Ntau, Dtau)$, respectively a `networkX` graph, $source$ and $target$ nodes, type of edge weight, maximum number of timesteps, and time resolution

**Output:** $(costs, paths)$, Two dictionaries keyed by node id: path costs from the source (e.g. cumulated $CO_2$ ), and corresponding optimal paths

1: $costs \leftarrow \{\}$

2: $seen \leftarrow \{source : 0\}$

3: $paths \leftarrow \{source : [source]\}$

4: *# fringe is a min-priority queue of $(cost, node)$ tuples*

5: $fringe \leftarrow heap()$

6: $fringe.push(0, source)$

7: **while** $fringe \neq \emptyset$ **do**

8:     $(d, v) \leftarrow fringe.pop()$

9:     **if** $v \in costs$ **then**

10:         *# Already visited node*

11:         $skip$

12:     **end if**

13:     $costs[v] \leftarrow d$

14:     **if** $v = target$ **and** $\forall n \in G.neigh(target), n \in seen$ **then**

15:         **exit**

16:     **end if**

17:     $t\_idx \leftarrow get\_time\_index(paths[v], d, wT, Ntau, Dtau)$

18:     *# Iterate on v's forward-star*

19:     **for** $(u, cost)$ in $G.succ(v)$ **do**

20:         *# evaluate edge weight of wT type at time step t_idx*

21:         $c \leftarrow cost.at\_time(t\_idx, wT)$

22:         $vu\_cost \leftarrow costs[v] + c$

23:         **if** $u \notin seen$ or $vu\_cost < seen[u]$ **then**

24:             $seen[u] \leftarrow vu\_cost$

25:             $fringe.push(vu\_cost, u)$

26:             $paths[u] \leftarrow paths[v] + [u]$

27:         **end if**

28:     **end for**

29: **end while**

**Algorithm 2** GET_TIME_INDEX

---

**Input:** $(paths, d, wT, Ntau, Dtau)$, respectively a dictionary of paths, node costs, type of edge weight, maximum number of timesteps, and time resolution

**Output:** $t\_idx$, the time step at which the costs $d$ are realised along the $paths$

1: **if** $wT =$ "*time*" **then**
2:     $t\_idx \leftarrow min(Ntau, \lfloor d/Dtau \rfloor)$
3: **else**
4:     *# compute cTime cumulative time*
5:     $cTime \leftarrow 0$
6:     $t\_idx \leftarrow 0$
7:     **for** $edge$ **in** $paths$ **do**
8:         *# evaluate edge delay at time step t_idx*
9:         $cTime \leftarrow cTime + edge.cost.at\_time(t\_idx, \text{"}time\text{"})$
10:        $t\_idx \leftarrow min(Ntau, \lfloor time/Dtau \rfloor)$
11:    **end for**
12: **end if**

---

determining a waiting time at the source node. In this case, waiting involves encountering more favourable edge delay values, leading to the computation of a faster path. In other words, employing a First-In-First-Out (FIFO) strategy for traversing graph
edges may not always be optimal. This is why it is referred to such a scenario as "non-FIFO". We observe that condition Eq. 21 is violated when a graph edge, which was initially unavailable for navigation, suddenly becomes accessible. While this is a rather infrequent event for motor vessels (in Mannarini and Carelli (2019b) non-FIFO edges were just $10^{-6}$ of all graph edges), it is a more common situation for sailboats navigating in areas where the wind is initially too weak or within the no-go zone (Eq. 7, Fig. 7). Indeed, the unavailability of an edge can be suddenly lifted as the wind strengthens or changes direction.
However, under a FIFO-hypothesis, the least-time algorithm would not wait for this improvement of the edge delay to occur. Rather, it would look for an alternative path avoiding the forbidden edge, potentially leading to a suboptimal path. In the case study of this paper, such a situation occurred for about $2 \cdot 10^{-3}$ of the total number of sailboat routes, see Sect. 5.2.2 and Supplement Sect. S3.2.

## 2.5   Vessel modeling

At the heart of the VISIR-2 kinematics of Sect. 2.1 are the vessel forward and transversal speed in a seaway, Eqs. 5a-5b. In what follows, such a vessel performance function is also termed as a "vessel model".

In VISIR-1 the forward speed resulted, for motor vessels, from a semi-empirical parametrisation of resistances (Mannarini et al., 2016a) and, for sailboats, from polar diagrams (Mannarini et al., 2015). The transversal speed due to leeway was neglected.

In VISIR-2 new vessel models were used, and just two of them are presented in this paper: a ferry and a sailboat. However, any other vessel type can be considered, provided that corresponding performance curve is utilised in the `Navi` module (Tab. 4, Fig. 8). The computational methods used to represent both the ferry and the sailboats are shortly described in Sect. 5.2.1-5.2.2. All methods provide the relevant kinematic quantities and, where applicable, the emission rates in correspondence of discrete values of the environmental variables. Such a "look-up table" (LUT) was then interpolated to provide VISIR-2 with a function

to be evaluated at the actual environmental (wave, currents, or wind) conditions. Additional LUT can be used as well, and the relevant function for this part of the processing is `vesselModels_identify.py`.

     The interpolating function either was a cubic spline (for sailboats) or the outcome of a neural network-based prediction scheme (for the ferry). The neural network features are provided in App. B. While the neural network generally demonstrated superior performance in fitting the LUTs (see Sect. S2 of the Supplement ), it provided unreliable data in extrapolation mode,

as shown in Fig. 6-7. In contrast, the spline, when extrapolation was requested, returned the value at the boundary of the input data range.

### 2.5.1   Ferry

The ferry modelled in VISIR-2 was a medium-size Ro-Pax vessel which parameters are reported in Tab. 2. A vessel's sea-keeping model was used at the ship simulator at the University of Zadar, as documented in Mannarini et al. (2021). Thereto,

additional details about both the simulator and the vessel can be found. The simulator applied a forcing from wind-waves of significant wave height $H_s$ related to the local wind intensity $V_i$ by

$$H_s[\text{m}] = 0.0055 \cdot V_i[\text{m/s}] + 0.0127 \cdot (V_i[\text{m/s}])^2 \tag{22}$$

This relationship was derived by Farkas et al. (2016) for the wave climate of the middle Adriatic Sea. The simulator then recorded the resulting vessel speed, as well as some propulsion and emission-related quantities. Leeway could not be considered

by the simulator. The post-processed data feeds the LUT to be then used for interpolating both the STW and $CO_2$ emission rate $\Gamma$ as functions of: significant wave height $H_s$, relative wind-wave direction $\delta_a = \delta_i$, and fractional engine load $\chi$. The results are displayed in Fig. 6.

     In a given sea state, the sustained speed is determined by the parameter $\chi$. For head seas ($\delta_a = 0°$) the STW is seen to decrease with $H_s$. The maximum speed loss varies from about 45% of the calm water speed at $\chi = 1$ to about 70% at $\chi = 0.7$

(Fig. 6.a). For $\chi = 0.7$, the STW sensitivity on $H_s$ decreases from head ($\delta_a = 0°$) to following seas ($\delta_a = 180°$, Fig. 6.b). For this specific vessel, the increase in roll motion in beam seas, as discussed in Guedes Soares (1990), and its subsequent impact on speed loss, does not appear to be a relevant factor.

     The $\Gamma$ rate, which is in the order of 1 $tCO_2$ per hour, shows a shallow dependence on $H_s$ (Fig. 6.c) while it is much more critically influenced by both $\chi$ and $\delta_a$ (Fig. 6.c.d).

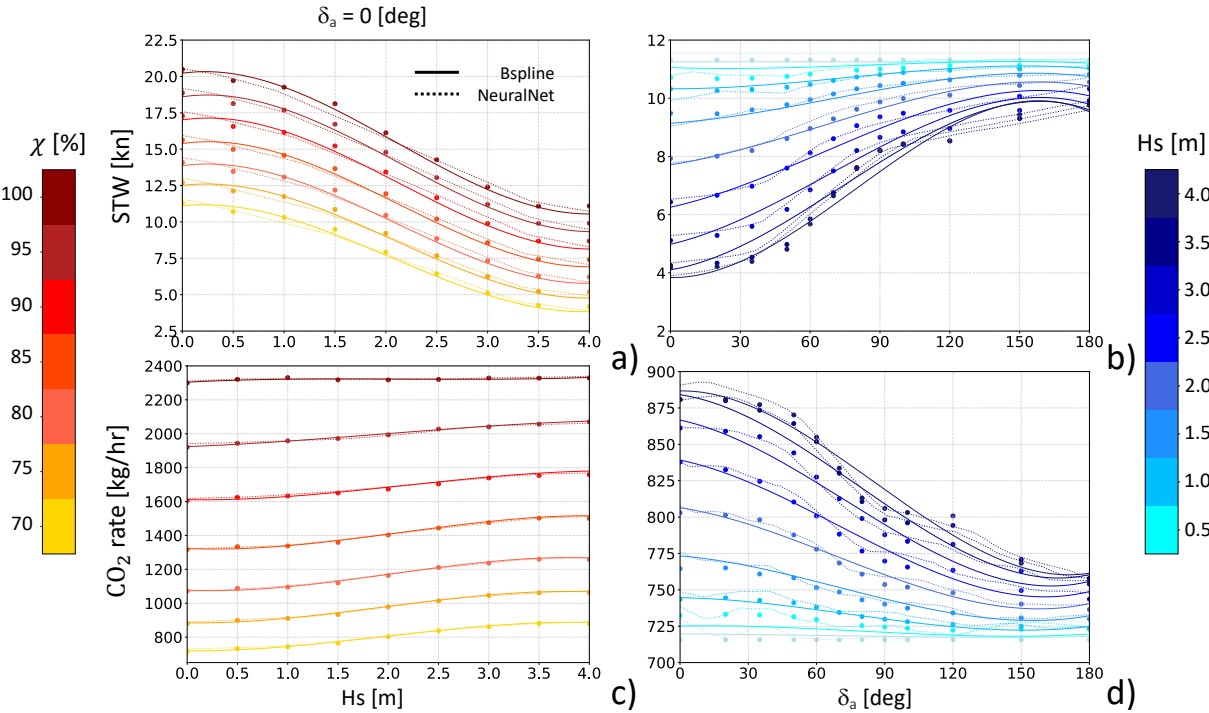

**Figure 6.** Ferry performance curve: In (a), STW is shown as a function of significant wave height $H_s$ for head seas ($\delta_a = 0°$), with engine load $\chi$ indicated by the marker color. In (b), STW is plotted as a function of $\delta_a$ at a constant $\chi = 0.7$, with $H_s$ represented by the colour variation. The lower panels (c, d) display the $CO_2$ emission rate ($\Gamma$) with similar dependencies as in panels (a, b). Markers correspond to the LUT values, solid lines represent the spline interpolation, and dotted lines indicate the neural network's output.

### 2.5.2 Sailboat

Any sailboat described in terms of polars can in principle be used by VISIR-2. For the sake of the case study, a Bénétau First-367 was considered. Its hull and rigging features are given in Tab. 3.

The modelling of the sailboat STW was carried out by means of the WinDesign Velocity Prediction Program (VPP). The tool was documented in Claughton (1999, 2003) and references therein. The VPP is able to perform a four degrees of freedom analysis, taking into account a wide range of semi-empirical hydrodynamic and aerodynamic models. It solves an equilibrium problem by a modified multi-dimensional Newton-Raphson iteration scheme. The analysis considered the added resistance due to waves by means the so-called "Delft method" based on the Delft Systematic Yacht Hull Series (DSYHS). Besides, the tool allows to introduce response amplitude operators derived from other techniques as well, such as computational fluid dynamics. The wind-wave relationship was assumed to be given by Eq. 22. For each wind configuration (i.e., speed and direction) the optimal choice of sails set was considered. The main sail and the jib sail were considered for upwind conditions, otherwise the combination of main sail and spinnaker was used.

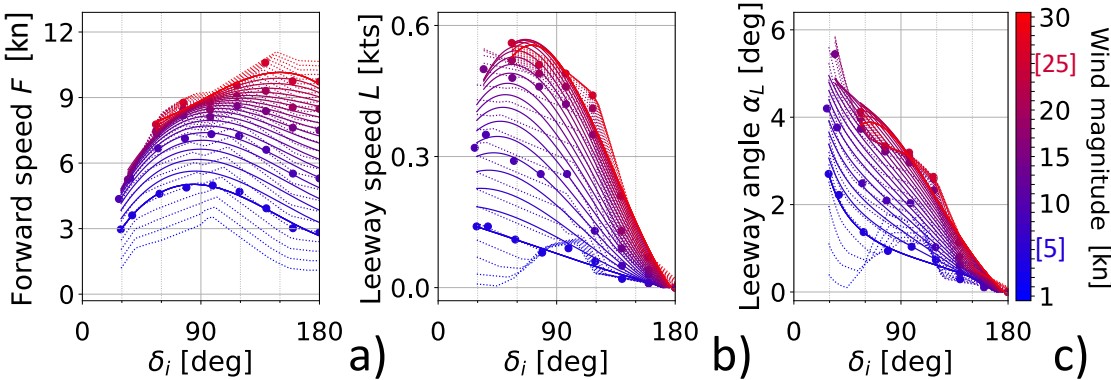

**Figure 7.** Sailboat performance curve: Forward speed $F$ in (a) and leeway speed $L$ in (b) are both plotted against the true wind angle $\delta_i$. Panel c) shows the leeway angle $\alpha_L$ obtained from Eq. 6. Marker and line colours represent wind magnitude $V_i$. Data start at $\delta_i = \alpha_0(V_i)$. Markers refer to the LUT, solid lines to spline interpolation, and dotted lines to the neural network's output. The colorbar also reports LUT's minimum and maximum values printed in blue and red, respectively.

The outcome corresponds to Eq. 5 and is provided in Fig. 7. The no-go angle $\alpha_0$ varies from 27 to 53°as the wind speed increases from 5 to 25 kn. At any true wind angle of attack $\delta_i$, the forward speed $F$ increases with wind intensity, especially at lower magnitudes (Fig. 7.a). The peak boat speed is attained for broad reach ($\delta_i \approx 135°$). Leeway magnitude $L$ instead is
475 at largest for points of sail between the no-go zone ($\delta_i = \alpha_0$) and beam reach ($\delta_i = 90°$), see Fig. 7.b. As the point of sail transitions from the no-go zone to running conditions, the leeway angle $\alpha_L$ gradually reduces from 6 to 0°. This decrease follows a roughly linear pattern, as depicted in Fig. 7.c.

## 2.6 Visualisation

Further innovations brought in by VISIR-2 regard the visualisation of the dynamic environmental fields and the use of isolines.
To provide dynamic information via a static picture, the fields are rendered via concentric shells originating at the departure location. The shape of these shells is defined by isochrones. These are lines joining all sea locations which can be reached from the origin, upon sailing for a given amount of time. This way, the field is portrayed at the time step the vessel is supposed to be at that location. Isochrones bulge along gradients of vessel's speed. Such shells represent an evolution of the stripe-wise rendering introduced in VISIR-1.b (Mannarini and Carelli, 2019b)[Fig.5]. The saved temporal dimensional of this plot type
allows for its application in creating movies, where each frame corresponds to varying values of another variable, such as the departure date or engine load, see the video supplement of this paper. This visual solution is another novelty introduced by VISIR-2.

In addition to isochrones, lines of equal distance from the route's origin (or: "isometres") and lines of equal amount of $CO_2$ emissions (or: "isopones") are also computed. The name isopone is related to energy consumption (the greek word
means "equal effort") which, for an internal combustion engine, the $CO_2$ emission is proportional to. Isopones bulge against

gradients of emissions. Isometres do not bulge, unless some obstruction (shoals, islands, landmass in general) prevents straight navigation. Given that rendering is on a Mercator map, "straight" refers to ship's movement along a constant bearing line. Isochrones correspond to the reachability fronts used in a model based on the Level Set Equation (LSE) by Lolla (2016).

## 2.7 Code modularity and portability

Software modularity has been greatly enhanced in VISIR-2. While in VISIR-1 modularity was limited to the graph preparation, which was detached from the main pipeline (Mannarini et al. (2016a)[Fig.8]), the VISIR-2 code is organised into many more software modules. Their characteristics are given in Tab. 4 while the overall model workflow is shown in Fig. 8. The modules

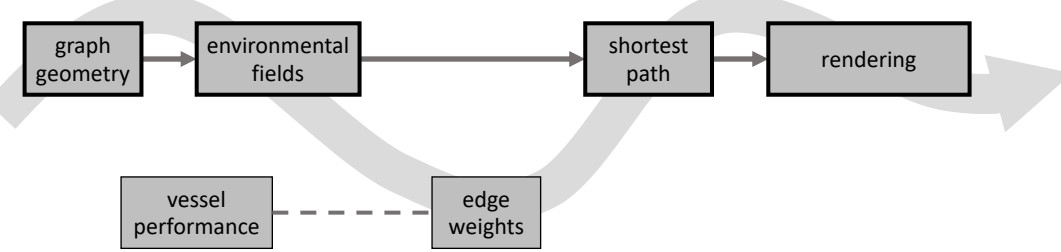

**Figure 8.** VISIR-2 workflow. Modules enclosed within thicker frames are intended for direct execution by the end-user, while the other modules can be modified for advanced usage. The data flow occurs along the wavy arrow, with routine calls along the dashed line.

can be run independently and can optionally save their outputs. Through the immediate availability of products from previously executed modules, this favours the research and development activities. For operational applications (such as GUTTA-VISIR)
instead, the computational workflow can be streamlined by avoiding the saving of the intermediate results. VISIR-2 module names are Italian words. This is done for enhancing their distinctive capacity, cf. Wilson et al. (2014). More details on the individual modules can be found in the user manual, provided as part of the present release (Salinas et al., 2024a).

A preliminary graphical user interface (GUI) is also available. In the hereby released version of VISIR-2, it facilitates the ports selection from the World Port Index[***] database.
VISIR-2 was developed on Mac OS Ventura (13.x). However, both path parameterisation and use of virtual environments ensure portability, which was successfully tested for both Ubuntu 22.04.1 LTS and Windows 11, both on personal computers and two distinct high-performance computing (HPC) facilities.

---

[***]https://msi.nga.mil/Publications/WPI

## 3 Validation

Validating a complex model like VISIR-2 is imperative. The code was developed with specific runs of VISIR-1 as a benchmark. The validation of VISIR-1 involved comparing its outcomes with both analytical and numerical benchmarks, and assessing its reliability through extensive utilization in operational services (Mannarini et al., 2016b).

Previous studies have compared VISIR-1 to analytical benchmarks for both static wave fields ("Cycloid", Mannarini et al. (2016a)) and dynamic currents ("Techy", Mannarini and Carelli (2019b)). Here, we present the results of executing the same tests with VISIR-2, at different graph resolutions, as shown in Table 5. The errors are consistently found to be below 1%. Additionally, in Mannarini et al. (2019b)[Tab. II], routes computed in dynamic wave fields were compared to the results from a model based on the LSE. For these specific runs of VISIR-2, the benchmarks were taken as LSE simulations at the nearest grid resolution. Notably, VISIR-2 consistently produces shorter-duration routes compared to VISIR-1.b (see Tab. 6). For both analytical and numerical benchmarks, distinct from the scenario discussed in Sect. 2.2.3, quasi-collinear edges were retained in the graphs.

During the tests mentioned earlier, the vessel's STW remained unaffected by vector fields. In instances where there was a presence of current vector fields ("Techy" oracle), they were merely added to the STW, without directly impacting it. Therefore, the enhanced capability of VISIR-2 to accommodate angle-dependent vessel performance curves (cf. Eq. 5) needs to be showcased.

To achieve this objective, the openCPN model was utilised. This model can calculate sailboat routes with or without factoring in currents and incorporates shoreline knowledge, though it does not consider bathymetry. For our tests, we provided VISIR-2 with the identical wind and sea current fields used by openCPN (further details are provided in Sect. 5.1). Additionally, both models were equipped with the same sailboat polars. However, it is worth noting that openCPN does not handle leeway, whereas VISIR-2 can manage it. The VISIR-2 routes were computed on graphs of variable mesh resolution $1/\Delta x$ and connectivity $\nu$, keeping fixed the "path resolution" parameter $\Delta P$ which was introduced in Mannarini et al. (2019b)[Eq.6]. This condition ensures that the maximum edge length remains approximately constant as the $(\Delta x, \nu)$ parameters are varied. Exemplary results are depicted in Fig. 9, with corresponding metrics provided in Tab. 7.

VISIR-2 routes exhibit topological similarity to openCPN routes, yet for upwind sailing, they require a larger amount of tacking (see Fig. 9.a). This discrepancy arises from the limited angular resolution of the graph (cf. Tab. 1). In the absence of currents, this implies a longer sailing time for VISIR-2 with respect to openCPN routes, ranging between 1.0 and 3.4%. However, as the graph resolution is increased, the route duration decreases. Notably, this reduction plateaus at $\nu = 7$, indicating that such a resolution is optimal for the given path length. This type of comparison addresses the suggestion by Zis et al. (2020) to investigate the role of resolution in weather routing models. For a more thorough discussion of this particular aspect, please refer to Mannarini et al. (2019b).

Downwind routes all divert Northwards because of stronger wind there (Fig. 9.b). For these routes, the angular resolution does not pose a limiting factor, and VISIR-2 routes exhibit shorter durations compared to openCPN routes. Considering the influence of currents as well, VISIR-2 routes consistently prove to be the faster option, even for upwind sailing.

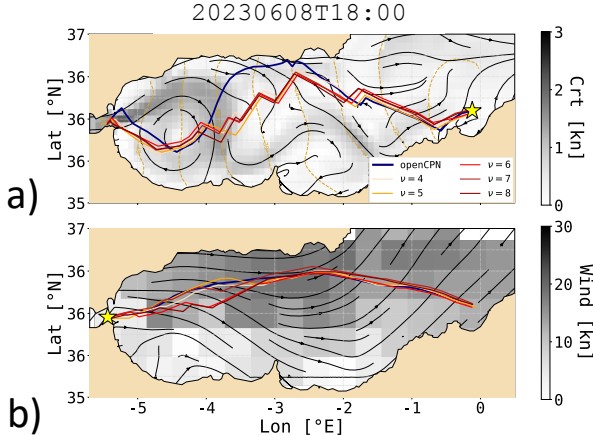

**Figure 9.** VISIR-2 routes with wind and currents vs. openCPN: Graphs of variable resolution, indexed by $\nu$ as shown in the legend, with a constant $\Delta P \sim 0.3°$. Field intensity is in grey tones, and the direction as black streamlines. Shell representation with isochrones in gold dashed lines and labels in $\mathrm{hr}$. The openCPN solution is plotted as a navy line. Panels a) and b) refer to the West- and Eastbound voyage, respectively.

The disparities in duration between openCPN and VISIR-2 routes could be attributed to various factors, including the interpolation method used for the wind field in both space and time, and the approach employed to consider currents. Delving into these aspects would necessitate a dedicated investigation, which is beyond the scope of this paper.

Numerical tests have been integrated into the current VISIR-2 release (Salinas et al., 2024a), covering the experiments listed in Tab.5-7 and beyond. These tests can be run using the `Validazioni` module.

## 4   Computational performance

The computational performance of VISIR-2 was evaluated using tests conducted on a single node of the "juno" HPC facility at CMCC. This node was equipped with an Intel Xeon Platinum 8360Y processor, featuring 36 cores, each operating at a clock
speed of 2.4 GHz, and boasting a per-node memory of 512 GB. Notably, parallelisation of the cores was not employed for these specific numerical experiments. Subsequently, our discussion here narrows down to assessing the performance of the module dedicated to computing optimal routes ("`Tracce`") in its motor vessel version.

In Fig. 10, we assess different variants of the shortest path algorithm: least-distance, least-time, and least-$CO_2$ . We differentiate between the core of these procedures, which focuses solely on computing the optimal sequence of graph nodes (referred
to hereafter as the "Dijkstra" component), and the broader procedure ("total"), which also includes the computation of both marine and vessel dynamical information along the legs of the optimal paths. The spatial interpolation option used in these tests (Sint = 1 of Sect. 2.3.2) provides a conservative estimation of computational performance.

The numerical tests utilise the number of degrees of freedom (DOF) for the shortest-path problem as the independent variable. This value is computed as $A \cdot N_\tau$, where $A$ denotes the number of edges, and $N_\tau$ stands for the number of time steps of the fine grid (cf. Fig. 4). In the context of a sea-only edges graph, particularly in the case of a large graph (where border effects can safely be neglected), $A$ can be represented as $4 \cdot N_{q1}(\nu)$, where $N_{q1}$ is defined by Eq. 20. Random edge weights were generated for graphs with $\nu = 10$, resulting in a number of DOF ranging between $10^5$ and $10^9$. Each data point in Fig. 10 represents the average of three identical runs, which helps reduce the impact of fluctuating HPC usage by other users. Additionally, the computational performance of VISIR-2 is compared to that of VISIR-1b, as documented in Mannarini and Carelli (2019b) [Tab.3, 'With T-interp'].

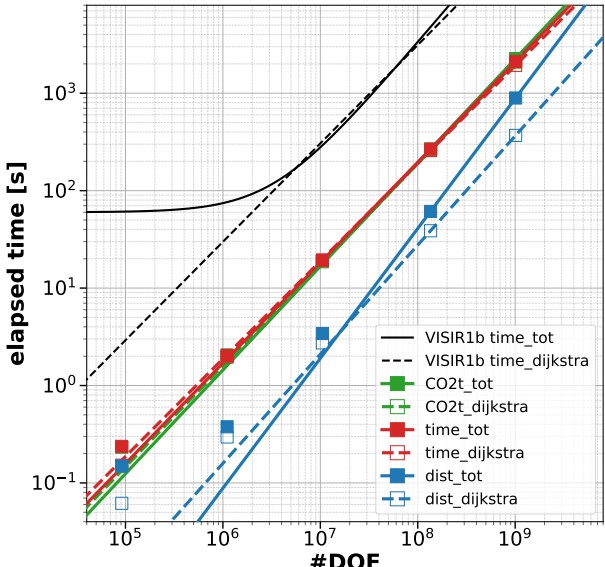

**Figure 10.** Profiling of computing time for the Tracce module (motor vessel case). The independent variable is the #DOF in the graph. Markers refer to experimental data points and lines to least-square fits. Void markers and dashed lines refer to just the Dijkstra's component, while full markers and solid lines to the whole routines. The colours refer to the three alternative optimisation objectives, while black is used for VISIR-1.b results.

The primary finding is a confirmation of a power-law performance for all three optimisation objectives of VISIR-2: distance, route duration, and total $CO_2$ emissions. Remarkably, the curves appear nearly linear for the latter two algorithms (see Tab. 8). Such a scaling is even better than the linear-logarithmic worst-case estimate for Dijkstra's algorithms (Bertsekas, 1998)[Sect.2.3.1]. Furthermore, this is not limited to just the Dijkstra components (as observed in VISIR-1.b), but extends to the entire procedure, encompassing the reconstruction of along-route variables. In addition to this enhanced scaling, VISIR-2 demonstrates an improved absolute computational performance within the explored DOF range. The performance gain is approximately a factor of 10 when compared to VISIR-1.b. This factor should even be larger when using the Sint = 0 interpolation option.

Digging deeper into the details, we observe that the least-distance procedure within VISIR-2, while exclusively dealing with static edge weights, exhibits a less favourable scaling behaviour, compared to both the least-time and least-$CO_2$ procedures. This is attributed to the post-processing phase, wherein along-route information has to be evaluated at the appropriate time step. Further development is needed to improve on this. Additionally, tiny non-linearities are seen for the smaller # DOF. However, as proven by the metrics reported in Tab. 8, they do not affect the overall goodness of the power-law fits.

Lastly, it was found that peak memory allocation scales linearly across the entire explored range, averaging about 420B per DOF. This is about five times larger than in VISIR-1b and should be attributed to the `networkX` structures used for graph representation. However, the large memory availability at the HPC facility prevented a possible degradation of performance for the largest numerical experiments due to memory swapping. A reduction of the unit memory allocation by a factor of two should be feasible using single precision floating point format. Another strategy would involve using alternative graph libraries, such as `igraph`.

A more comprehensive outcome of the VISIR-2 code profiling, distinguishing also between the sailboat and the motor vessel version of the `Tracce` module, is provided in the S1 section of the Supplement .

## 5   Case studies

A prior version of VISIR-2 has empowered both GUTTA-VISIR operational service, generating several million optimal routes within the Adriatic Sea over the span of a couple of years. In this section, we delve into outcomes stemming from deploying VISIR-2 in different European seas. While the environmental fields are elaborated upon in Sect. 5.1, the results are given in Sect. 5.2, distinguishing by ferry and sailboat.

### 5.1   Environmental fields

The fields used for the case studies include both static and dynamic fields. The only static one was the bathymetry, extracted from the EMODnet product of 2020[†††]. Its spatial resolution was 1/16 arcmin. The dynamic fields were the metocean conditions from both the European Centre for Medium-Range Weather Forecasts (ECMWF) and CMEMS. Analysis fields from the ECMWF high resolution Atmospheric Model, 10-day forecast (Set I - HRES) with 0.1° resolution[‡‡‡] were obtained. Both the u10m and v10m variables with six-hourly resolution were used. From CMEMS, analyses of the sea state, corresponding to the hourly MEDSEA_ANALYSISFORECAST_WAV_006_017 product, and of the sea surface circulation, hourly MEDSEA_ANALYSISFORECAST_PHY_006_013, both with 1/24° spatial resolution, were obtained. The wind-wave fields (vhm0_ww, vmdr_ww) and the Cartesian components (uo,vo) of the sea surface currents were used, respectively. Just for the comparison of VISIR-2 to openCPN (see Sect. 3), three-hourly forecast fields from ECMWF (0.4°, [§§§]) and three-hourly Real-Time Ocean Forecast System (RTOFS) forecasts (1/12°, https://polar.ncep.noaa.gov/global/about/) for surface currents

---

[†††]https://emodnet.ec.europa.eu/en/bathymetry
[‡‡‡]https://www.ecmwf.int/en/forecasts/datasets/set-i#I-i-a_fc
[§§§]https://www.ecmwf.int/en/forecasts/datasets/open-data

(p3049 and p3050 variables for U and V respectively) were used, respectively. Time resolution was three-hourly for both products.

The kinematics of VISIR-2 presented in Sect. 2.1 do not limit the use to just surface ocean currents. This was just an initial approximation based on the literature discussed in Mannarini and Carelli (2019b). However, recent multi-sensor observations reported in Laxague et al. (2018) at a specific location in the Gulf of Mexico revealed a significant vertical shear, both in magnitude (by a factor of 2) and direction (by about 90 degrees), within the first 8 m. Numerical ocean models typically resolve this layer, for instance the mentioned Mediterranean product of CMEMS provides four levels within that depth. This

vertically resolved data holds the potential to refine the computation of a ship's advection by the ocean flow. A plausible approach could involve the linear superposition of vessel velocity with a weighted-average of the current, considering also the ship's hull geometry.

## 5.2   Results

To showcase some of the novel features of VISIR-2, we present the outcomes of numerical experiments for both a ferry (as

outlined in Sect. 5.2.1) and a sailboat (Sect. 5.2.2). All the results were generated using the interpolation options Sint=0 (as elaborated upon in Sect. 2.3.2) and Tint=2 (Sect. 2.3.1). These experiments considered the marine and atmospheric conditions prevailing in the Mediterranean Sea during the year 2022. Departures were scheduled daily at 03:00 UTC. The percentage savings ($dQ$) of a given quantity $Q$ (such as the total $CO_2$ emissions throughout the journey or the duration of sailing) are computed comparing the optimal ("opt") to the least-distance route ("gdt"):

$$dQ = \frac{Q^{(\mathrm{opt})} - Q^{(\mathrm{gdt})}}{Q^{(\mathrm{gdt})}} \qquad (23)$$

### 5.2.1   Ferry

The chosen domain lies at the border between the Provençal Basin and the Ligurian Sea. Its sea state is significantly influenced by the Mistral, a cold northwesterly wind that eventually affects much of the Mediterranean region during the winter months. The circulation within the domain is characterized by the southwest-bound Liguro-Provençal current and the associated eddies.

(Schroeder and Chiggiato, 2022).

We conducted numerical experiments using VISIR-2 with a graph resolution given by $(\nu, 1/\Delta x) = (4, 12/°)$, resulting in 2,768 nodes and 114,836 edges within the selected domain. The time grid resolution was set at $\Delta\tau = 30\mathrm{min}$ and $N_\tau = 40$. A single iteration ($k = 1$) of equation Eq. A1 was performed. The ferry engine load factor $\chi$ was varied to encompass values of 70, 80, 90, and 100% of the installed engine power. For each day, both route orientations, with and without considering

currents, were taken into account. This led to a total of 5,840 numerical experiments. The computation time for each route was approximately 4 min, with the edge weight and shortest path calculations consuming around 30 sec.

In Fig. 11.a an illustrative route is shown during a Mistral event. As the ferry navigates against the wind, both its speed loss and $CO_2$ emission rate reach their maximum levels (cf. Fig. 6.b.d). Consequently, both the least-time and the least-$CO_2$ algorithms calculate a detour into a calmer sea region where the combined benefits of improved sustained speed and reduced

$CO_2$ emissions compensate for the longer path's costs. The least-$CO_2$ detour is wider than the least-time one, as the additional duration is compensated by greater $CO_2$ savings. Moreover, these detours maximize the benefits from a southbound meander of the Liguro-Provençal current. Both optimal solutions intersect the water flow at points where it is narrowest, minimising the speed loss caused by the crosscurrent (cf. Eq. 16). Recessions of the isochrones become apparent starting at 6 hr since departure. For this specific departure date and time, the overall reduction in $CO_2$ emissions, in comparison to the shortest-

distance route, exceeds 35%.

     Fig. 11.b illustrates that the magnitude of the related spatial diversion is merely intermediate, compared to the rest of 2022. Particularly during the winter months, the prevailing diversion is seen to occur towards the Ligurian Sea. Notably, VISIR-2 even computed a diversion to the East of Corsica, which is documented in Sect. S3.1 of the Supplement . In the supplementary video accompanying this manuscript, all the 2022 routes between Port Torres and Toulon are rendered, along with relevant

environmental data fields.

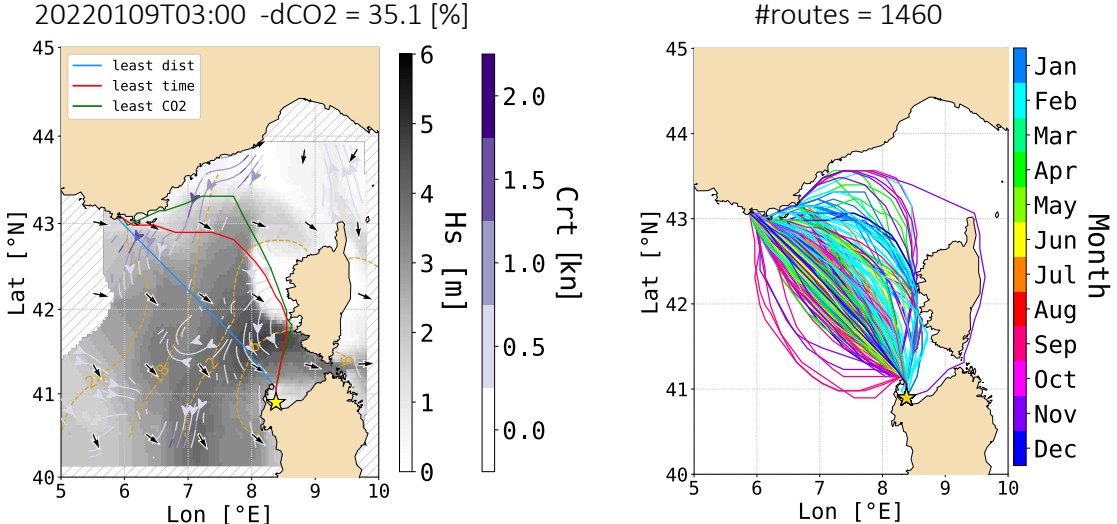

**Figure 11.** Ferry's optimal routes between ITPTO and FRTLN with both waves and currents: a) For the specified departure date and time, the shortest-distance route is shown in blue, the least-time route in red, and the least-$CO_2$ route in green. The $H_s$ field is displayed in shades of grey with black arrows, while the currents are depicted in purple tones with white streamlines. The shortest path algorithm did not utilise environmental field values within the etched area. Additionally, isochrones of the $CO_2$ -optimal route are shown at 3-hourly intervals. The engine load was $\chi = 0.7$. b) A bundle of all northbound $CO_2$ -optimal routes (for $\chi = [0.7, 0.8, 0.9, 1.0]$) is presented, with the line colour indicating the departure month.

     To delve deeper into the statistical distribution of percentage $CO_2$ savings defined as in Eq. 23, Fig. 12.a provides a comparison with both the average significant wave height $\langle H_s^{(\mathrm{gdt})} \rangle$ and absolute wave angle of attack $\langle |\delta_a^{(\mathrm{gdt})}| \rangle$ along the shortest-distance route. Firstly, it should be noted that an increase in wave height can lead to either substantial or minimal $CO_2$ emission savings. This outcome depends on whether the prevailing wave direction is opposing or aligned with the vessel's heading. When

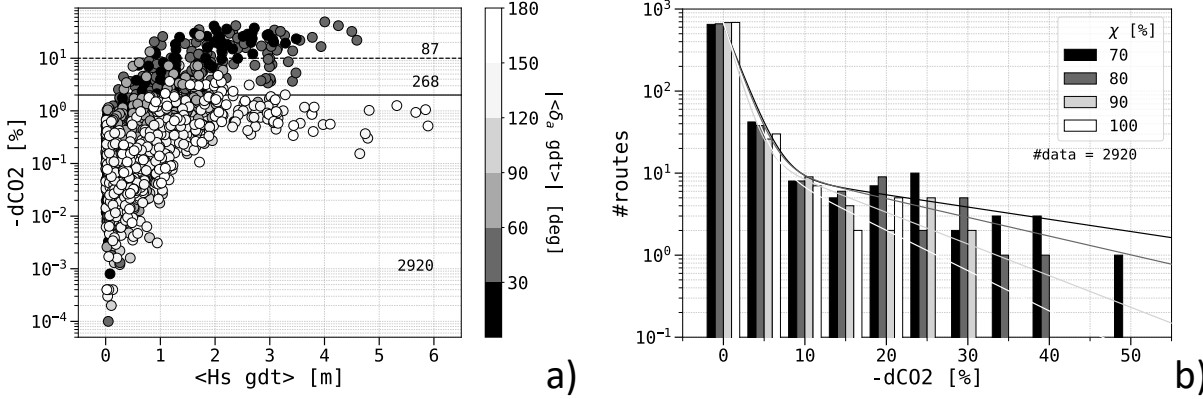

**Figure 12.** Metrics relative to ferry routes pooled on sailing directions (FRTLN ↔ ITPTO) and $\chi$, using both waves and currents. a) Percentage savings, with marker's grey shade representing the mean angle of attack along the least-distance route. The total number of routes, those with relative $CO_2$ savings above 2% (solid line) and 10% (dashed), are also provided; b) Distributions of the $CO_2$ savings for each $\chi$ value, with fitted bi-exponential functions as in Tab. 10. Each set of four columns pertains to a bin centred on the nearest tick mark and spanning a width of 5%.

focusing on routes with a $CO_2$ saving of at least 2%, it is seen that they mostly refer to either beam or head seas along the least-distance route. This corresponds to elevated speed loss and subsequent higher emissions, as reported in Fig. 6b.d. This subset of routes shows a trend of larger savings in rougher sea states. Conversely, when encountering following seas with even higher $H_s$, savings remain below 1%. This is due to both a smaller speed reduction and a lower $CO_2$ emission rate. The counts of routes surpassing the 2% saving threshold accounts for more than one-tenth of the total routes, the ones above the

10% threshold, represent about 1/38th of the cases. This implies that, for the given ferry and the specified route, double-digit percentage savings can be anticipated for about ten calendar days per year.

The analysis of the $CO_2$ savings distribution can be conducted by also considering the role of the engine load factor $\chi$, as depicted in Fig. 12.b. The distribution curves exhibit a bi-exponential shape, with the larger of the two decay lengths ($d_2$) inversely proportional to the magnitude of $\chi$, cf. Tab. 10. This relationship is connected to the observation of reduced speed

loss at higher $\chi$ as rougher sea conditions are experienced, which was already noted in the characteristics of this vessel in Sect. 5.2.1. The distribution's tail can extend to values ranging between 20 and 50%, depending on the specific value of $\chi$.

Percentage $CO_2$ savings, broken down by sailing direction and considering the presence or absence of currents, are detailed in Tab. 9. The average savings range from 0.6% (1.0% when considering sea currents) to 1.9% (2.2%). It is confirmed that the savings are more substantial on the route that navigates against the Mistral wind (from Porto Torres to Toulon). However, the

665 percentage savings amplify when currents are taken into account, and this effect is particularly noticeable for the routes sailing in the downwind direction.

Further comments regarding the comparison of the $CO_2$ savings presented here with the existing literature can be found in in Sect. 6. While other works also hint at the presence of optimal route bundles, VISIR-2 marks the first comprehensive exploration of how $CO_2$ savings are distributed across various environmental conditions and engine loads.

### 5.2.2 Sailboat

The chosen area lies in the southern Aegean Sea, along a route connecting Greece (Monemvasia) and Turkey (Marmaris). This area spans through one of the most archipelagic zones in the Mediterranean Sea, holding historical significance as the birthplace of the term "archipelago". The sea conditions in this area are influenced by the Meltemi, a prevailing northerly wind, particularly during the summer season. Such an "Etesian" weather pattern can extend its influence across a substantial portion of the Levantine basin (Lionello et al., 2008; Schroeder and Chiggiato, 2022). On the eastern side of the domain, the circulation is characterised by the westbound Asia Minor Current, while on its western flank, two prominent cyclonic structures separated by the West-Cretan anticyclonic gyre are usually found. (Theocharis et al., 1999).

We performed numerical experiments with VISIR-2, with a graph resolution of $(\nu, 1/\Delta x) = (5, 15/°)$, leading to 2,874 nodes and 156,162 edges in the selected domain. The resolution of the time grid was $\Delta \tau = 30$ min. Furthermore, $N_\tau = 120$ time steps of the environmental fields and $k = 2$ iterations for Eq. A1 were used. A First-367 sailboat was selected. For each day, both route orientations, and all possible combinations of wind, current, and leeway were considered. This implied a total of 2,920 numerical experiments. Each route required a total computing time of about 7 min, of which the edge weight and shortest path computation amounted to 4 min, mainly spent in the edge weight computation. The excess time in comparison to the motor vessel's case study is attributed to both a higher value of $N_\tau$ and the additional time required for accounting for the exclusion of the no-go zone of the sailboat shown in Fig. 7.

In Fig. 13.a, a sailboat route is depicted for a specific departure date, superimposed on the wind and sea current patterns. Both the least-distance and least-time routes appropriately steer clear of continental or insular landmasses, with the time-optimal route opting for a more extensive detour. This adjustment is aimed at harnessing more favourable winds and circumventing unfavourable or cross currents, culminating in a remarkable 14.6% reduction in route duration.

Moving to Fig. 13.b, the collective set or "bundle" of eastbound routes is presented. Unlike the ferry routes showcased in Fig. 11.b, it proves more challenging to discern a distinct seasonal pattern for the diversions of the sailboat routes. A metric for measuring diversions, such as the Fréchet distance utilised in Mannarini et al. (2019a), could facilitate the identification of patterns. The corresponding return routes are shown in Sect. S4.2 of the Supplement , confirming this trend. The bundles indicate that accounting also for currents leads to a more expansive set of optimal routes.

In only five cases (constituting $1.7 \times 10^{-3}$ of all sailboat routes), the least-time route was discovered to be slower than the least-distance route. These instances are scrutinised in Sect. S3.2 of the Supplement . The apparent inconsistency arises from the fact that these least-time routes arrive at certain intermediate waypoints earlier, but encounter less favourable sailing conditions compared to those encountered by the least-distance routes arriving later. This discrepancy points to a non-FIFO situation (refer to Sect. 2.4.1). This scenario necessitates advancements in the least-time algorithm to accommodate dynamic edge weights, potentially incorporating an initial waiting time, as discussed in Orda and Rom (1990).

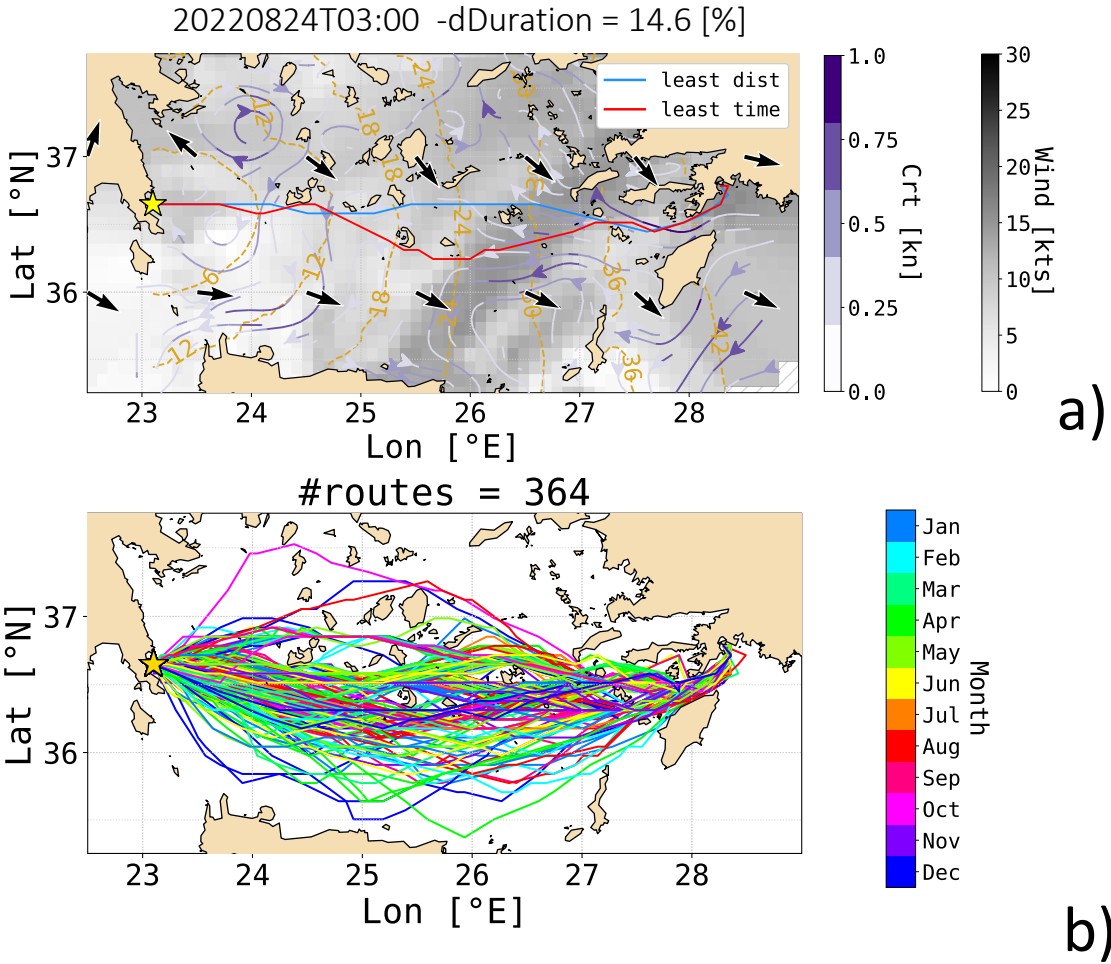

**Figure 13.** Sailboat's optimal routes between GRMON and TRMRM, considering both wind and currents: a) For the specified departure date and time, the least-time route is depicted in red, and the shortest-distance one in blue. The wind field is represented in shades of grey with black arrows, while the currents are shown in purple tones with white streamlines. Additionally, isochrones of the time-optimal route are displayed at 6-hourly intervals. b) A bundle of all eastbound time-optimal routes is presented, with the line colour indicating the departure month.

A statistical evaluation of the time savings resulting from the optimisation process for sailboat routes is illustrated in Fig. 14.a. Thereto, Eq. 23 is employed to assess both the path length and duration percentage savings. While the $-dT$ savings are generally proportional to the path lengthening $dL$, the most substantial savings manifest under nearly upwind conditions along the least-distance route, i.e. where $\langle |\delta_i^{(\mathrm{gdt})}| \rangle \sim \alpha_0$ (cf. Eq. 7). This is understandable, as reduced sustained speeds and extended edge sailing times occur when wind originates from sectors close to the no-go zone, as depicted in Fig. 7.a. However, it is worth noting that, under excessively weak or consistently sustained upwind conditions, a sailboat route might become unfeasible. A quantitative overview of such "failed" routes is provided in Tab. 11. It is evident that, thanks to the spatial diversions introduced by the route optimisation process, the likelihood of a least-time route failing, compared to the least-distance one, is reduced by a factor of approximately 100. In the video supplement accompanying this paper, all sailboat routes between Monemvasia and Marmaris in 2022, along with the corresponding environmental fields, are included.

In Fig. 14.b of the impact of currents and leeway is assessed. The influence of currents leads to a change in duration of up to approximately 5% when compared to routes affected solely by the wind. Categorising the data based on sailing direction (as presented in Sect. S5 in Supplement ), currents primarily contribute to shorter route durations for westbound courses (benefiting from the Asia Minor current). Conversely, they primarily result in extended durations for eastbound routes, particularly where, to the north of the island of Rhodes, there is no alternative but to sail against the current.

Turning to leeway, when not in combination with currents, it consistently extends the duration of routes. Particularly, as indicated in the Supplement , when facing upwind conditions (more likely for westbound routes), as the speed loss is exacerbated due to a higher leeway speed (region with $\delta_i \sim \alpha_0$ in Fig. 7.b).

As in our earlier comment in Sect. 2.1, the impact of leeway is mainly provided by its cross-course component, which invariably decreases the vessel's SOG. Notably, the longitudinal component is smaller than the cross-one by a $\tan \delta$ factor, cf. Eq. 17. With $\delta$ estimated from Eq. 14 and Fig. 7.b to fall within a range of a few degrees, the along-edge projection of leeway, $w_\parallel^{(L)}$, measures approximately one-tenth of the transversal one, $w_\perp^{(L)}$.

When both effects, currents and leeway, are considered together, the distribution of duration changes in comparison to wind-only routes resembles the distribution for the case with currents only. However, due to the impact of leeway, it is slightly skewed towards longer durations.

Finally in Tab. 11 time savings averaged throughout the year are presented. These savings are further categorised based on the direction of sailing and the specific combination of effects, including wind, currents, and leeway. The impact of sea currents generally increases the percentage duration savings from 2.4% (the directional average of the wind-only or wind and leeway cases) to 3.2% (the average across all results affected by sea currents). We observe that routes featuring prevailing upwind conditions and favourable westbound currents, while also accounting for leeway, typically yield greater percentage duration savings compared to corresponding eastbound routes. This outcome can be attributed to the increase in duration of the least-distance route, which results from the loss of sailboat manoeuvrability as the nogo-zone is approached. It is also noted that the number of failed routes increases in the presence of leeway during upwind sailing. More statistical metrics are provided in Tab.S9 of the Supplement . Finally we observe that, in VISIR-2, rigging is regarded as fixed (cf. Tab. 3), whereas in actual sailing practice, it may be optimised based on both the wind intensity and the point of sail.

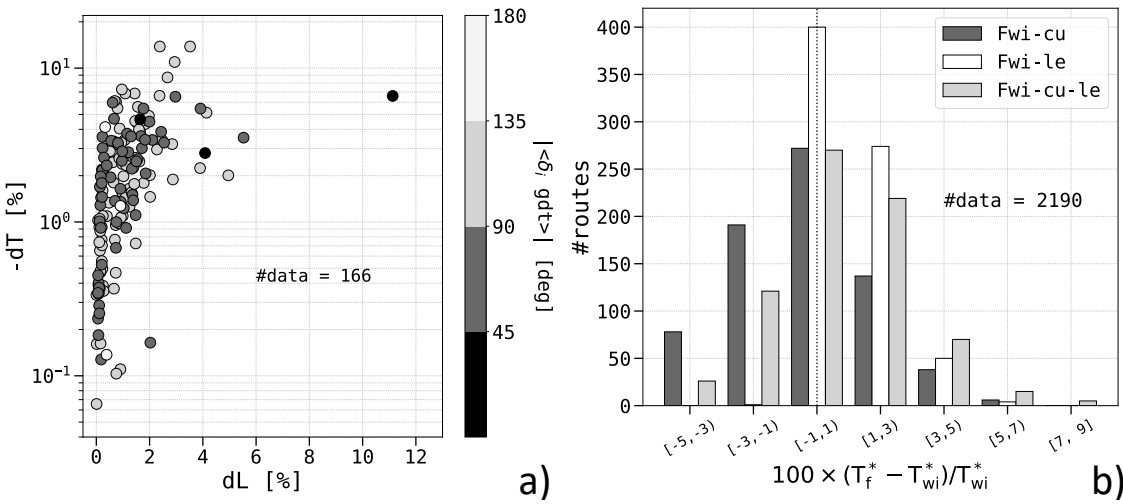

**Figure 14.** Metrics of the sailboat's optimal routes pooled on sailing directions (GRMON ↔ TRMRM). a) Duration percentage savings $-dT$ vs. relative lengthening $dL$ considering just wind: the marker's grey shade represents the average angle of attack of wind $|\langle \delta_i^{(\text{gdt})} \rangle|$ along the least-distance route. The #data is given by $\sum_d 365 - N_f^{(g)}$ where $d$ are the two sailing directions and the $N_f^{(g)}$ values are from the first row in Tab. 11. b) Histograms of relative route duration $T_f$, with environmental forcing combination $f$ defined by the column colour, with respect to the duration $T_{\text{wi}}$ of the wind-only optimal routes.

## 6 Discussion

In this section, we critically compare the $CO_2$ savings achieved in Sect. 5.2.1 for the ferry's optimal routes with those reported in the ship weather routing literature (Sect. 6.1). Furthermore, we explore potential applications and users of VISIR-2 (Sect. 6.2), and provide some insights of possible developments (Sect. 6.3).

### 6.1 Comparing CO₂ savings with literature

Only a handful of peer-reviewed papers have reported results on emission savings through ship routing, yet a few of these findings are reviewed here for comparison with VISIR-2's results.

For the ferry case study examined in this manuscript, the $CO_2$ emissions can be halved compared to those along the least-distance route, in the best-case (Fig. 12). This is in numerical agreement with the broad (0.1 - 48)% range reported in Bouman et al. (2017)[Tab.2]. Notably, the upper limit reduction stands as an outlier, with the more probable values depicted in Bouman et al. (2017)[Fig.2] falling below 10%. This aligns well with the results obtained from the statistical distribution of VISIR-2 experiments, presented in both Fig. 12.b and Tab. 9 of this manuscript.

Applying the VOIDS model, Mason et al. (2023a)[Sect.3.2] discovered that, for eastbound routes of a Panamax bulk carrier in the North Atlantic, voyage optimisation contributed to carbon savings ranging from 2.2 to 13.2%. This is a narrower range

compared to the present findings of VISIR-2 (cf. Fig. 12). However, both a different vessel type (a ferry) and a different domain of sailing (the Western Mediterranean Sea) were considered in our numerical experiments.

Miola et al. (2011) presented data from the second IMO GHG study, where the estimated $CO_2$ abatement potential for weather routing on the emissions of 2020 was reported to be as low as 0.24%. In the same paper, also a DNV study on projected emissions in 2030 was cited, providing an estimate of 3.9% for the $CO_2$ abatement potential through weather routing.

The former figure compares well to the average emission reduction computed via VISIR-2 for the ferry downwind conditions and high engine load, the latter to results for upwind and low engine load (cf. Tab. 9).

Lindstad et al. (2013) estimated the reduction in $CO_2$ emissions for a dry bulk Panamax vessel navigating in head seas during a typical stormy period in the North Atlantic. This reduction was determined when sailing on a 4,500 nautical miles (nmi) route compared to the shorter (yet stormier) least-distance route of 3,600 nmi. They found reductions ranging from 11

to 48%, depending on the speed of the vessel.

We note that, as e.g. in Mason et al. (2023a)[Fig.7], also VISIR-2 optimal routes exhibit spatial variations contingent on the departure date, forming a "bundle" as illustrated in Fig. 11.b and Sect.S4 of the Supplement . The shape of route boundaries was assessed for the United States' Atlantic coast routes by means of AIS data in Breithaupt et al. (2017). However, while multimodal distributions depending on sailing direction were noted, the authors did not attribute the preferential lanes to the

765 presence of ocean currents, but speculated that it was due to bathymetric constraints or artificial aids to navigation.

VISIR possesses a capability to incorporate ocean currents into the voyage optimisation process. As shown in Mannarini and Carelli (2019b), this integration has proven to significantly reduce the duration of transatlantic routes. In the present manuscript, we reaffirm the positive impact of currents on ship route optimisation, extending their benefits also to the reduction of $CO_2$ emissions (Tab. 9) and to the determination of more faithful duration savings for sailboat routes (Tab. 11).

In general, both average and extreme $CO_2$ emission percentage savings found in literature align well with the results obtained in the ferry case study presented in our manuscript. Nevertheless, engaging in a meaningful discussion of numerical differences, given the diverse range of vessel types, routes, environmental fields, and computational methods employed in the various published case studies, proves challenging.

VISIR-2 contributes to the existing body of literature by providing an open computational platform that facilitates the

775 simulation of optimal ship routes in presence of waves, currents, and wind. These simulations are designed to be transparent, with customisable sea domain and vessel performance curves, allowing for thorough inspection, modification, and evaluation. This addresses the concern raised by Zis et al. (2020) regarding the necessity of benchmarking instances of optimal routes and the associated input data. By providing such benchmarks, VISIR-2 supports and streamlines the work of future researchers in the field. Hence, we believe that the critical task of evaluating inter-model differences will best be addressed through dedicated

inter-comparison studies, as previously demonstrated with VISIR-1 (Mannarini et al., 2019b).

## 6.2 Potential uses of VISIR-2

Given its open-source nature, validated results, and numerical stability, VISIR-2 can hold great utility across various fields. The fact that both motor vessels and sailboats are treated equally will make VISIR-2 suitable for use in weather routing of vessels with wind-assisted propulsion.

Moreover, VISIR-2 can serve as a valuable tool for regulatory bodies seeking to inform policies on shipping. As previously outlined, agencies like EMSA, which oversee systems for monitoring, reporting, and verifying emissions, could utilise the model –provided that vessel performance curves and GHG emission rates are available– to calculate baseline emissions for various vessel- and GHG types. With the inclusion of shipping into the EU-ETS, shipowners, ship managers, or bareboat charterers –whoever bears the fuel cost– are mandated to surrender allowances for their emissions. These stakeholders may find it beneficial to explore open-source solutions alongside existing commercial options.

VISIR-2 also has the potential to help reduce uncertainty regarding the effectiveness of weather routing in reducing $CO_2$ emissions (Bullock et al., 2020). For instance, evaluating distributions as in Fig. 12.b, it becomes possible to characterise the joint potential of sea domains and vessel types for GHG emission savings. This concept also aligns with the idea of a "green corridor of shipping", as envisioned by both the Clydebank Declaration (gov.uk, 2021) and the United States' Department of State (DoS, 2022). In these initiatives, VISIR-2, thanks to the generality of its optimisation algorithm (Alg. 1, Alg. 2), could play a crucial role in minimising the consumption of costly zero-carbon fuel.

Furthermore, VISIR-2 could be used to generate a dataset of optimal routes for the training of artificial intelligence systems for autonomous vessels (Li and Yang, 2023), surpassing the shortcomings of using AIS tracks, which include incomplete coverage (Filipiak et al., 2020). Finally, we note that, as an open-source software, VISIR-2 can even have educational purposes, providing training opportunities for ship officials and maritime surveillance authorities, as well as for beginner sailors.

## 6.3 Outlook

There are several possible avenues for future developments of VISIR-2: they attain the computer science, the algorithms, the ocean engineering, and the environmental fields.

First, as mentioned in Sect. 4, some computational performance improvements for the least-distance procedure should be feasible. In applications where large domains, hyper-resolution, or multiple input environmental fields are required, it will be necessary to devise a solution that effectively reduces the computer's memory allocation. To further enhance modularity of VISIR-2, future developments can focus on object-oriented programming principles. Containerisation of VISIR-2 is currently underway as part of the development of a digital twin of the ocean (https://www.edito-modellab.eu/).

Further algorithmic work could address, for instance: construction of the graph directly in the projection space, facilitating the presence of more isotropic ship courses and perfectly collinear edges (cf. Sect. 2.2.2); development of an algorithm for given-duration, least-$CO_2$ routes; incorporation of multi-objective optimisation techniques (Szlapczynska, 2015; Sidoti et al., 2017); generalisation of the least-time algorithm to non-FIFO situations (cf. Sect. 2.4.1); consideration of tacking time and motor-assistance for sailboats or WAPS.

Transitioning to upgrades in ocean engineering for VISIR-2, the focus could shift towards targeting large ocean-going vessels, contingent upon the availability of related performance curves. Safety constraints on vessel intact stability (IMO, 2018b), considerations for slamming, green water, and lateral acceleration (Vettor and Guedes Soares, 2016), and passenger comfort as highlighted by Carchen et al. (2021), or vessel performance in the presence of cross-seas could be integrated. Second generation intact stability criteria (Begovic et al., 2023) could be accounted for through a dynamic masking of the graph. VISIR-2's readiness for wind-assisted ship propulsion hinges on the availability of an appropriate vessel performance curve that accounts for the added resistance from both wind and waves.

In terms of environmental data, it should be feasible to extend beyond the reliance on surface currents alone by incorporating the initial few depth layers of ocean models. Furthermore, VISIR-2 currently operates under the assumption of having perfect knowledge of meteo-oceanographic conditions, which is provided through forecast fields for shorter voyages or analysis fields for longer ones. The latter corresponds to retracked routes as discussed in Mason et al. (2023b). However, for real-time applications during extended voyages, it is essential to incorporate adaptive routing strategies. This entails using the latest forecasts to execute re-routing as needed.

## 7   Conclusions

This manuscript presented the development of VISIR-2: a modular, validated, and portable Python-coded model for ship weather routing. It provides a consistent framework for both motor vessel and sailboats by accounting for dynamic environmental fields such as waves, currents, and wind. The model can compute optimal ship routes even in complex and archipelagic domains. A cartographic projection, a so far overlooked feature, has been introduced in VISIR-2. The model provides, for vessels with an angle-dependent performance curve, an improved level of accuracy in the velocity composition with sea currents. It is found that heading and course differ by an angle of attack, which is given by the solution of a transcendental equation (Eq. 13) involving an effective flow being the vector sum of currents and leeway. A computationally inexpensive iterative solution has been devised (App. A). Furthermore, a variant of the Dijkstra's algorithm is introduced and used, which can minimise not just the $CO_2$ emissions but any figure of merit depending on dynamic edge weights (cf. Alg. 1, Alg. 2).

The validation of VISIR-2 included comparisons to oracles and two inter-comparison exercises. Different from the few available ship weather routing packages or services, the VISIR-2 software is accompanied by comprehensive documentation, making it suitable for community use.

The computational performance of the VISIR-2 shortest path module displayed a significant enhancement compared to its predecessor, VISIR-1 (Sect. 4). A quasi-linear scaling with problem complexity was demonstrated up to one billion DOF. The robustness of VISIR-2 was demonstrated across thousands of flawless route computations.

While the model is general with respect to vessel type, two case studies with VISIR-2, based on realistic vessel seakeeping models, were documented in this paper.

From nearly six thousand routes of a 125-meter-long ferry, computed considering both waves and currents in the North-Western Mediterranean, average $CO_2$ savings between 0.6 and 2.2%, depending on engine load and prevailing sailing direction,

were found. The distribution of the savings was bi-exponential, with the longer decay length becoming more pronounced at lower engine loads. This implied in particular that two-digit percentage $CO_2$ savings, up to 49%, were possible for about ten days annually. This statistical distribution sheds new light on the underlying factors contributing to the variability observed in the role of weather routing, as reported in previous review studies (Bouman et al., 2017; Bullock et al., 2020). Furthermore, our findings bear significance for both the environmental impact of greenhouse gas emissions and the financial considerations within the EU-ETS.

From close to three thousand routes of an 11-meter sailboat, within the Southern Aegean Sea, accounting for both wind and currents, an average sailing time reduction of approximately 2.5% was observed. When considering currents as a factor, the duration of optimal routes could further be reduced to 3%. Additionally, confirming prior work by Sidoti et al. (2023), disregarding the role of leeway would lead to an incorrect estimation of the route duration in upwind conditions. Several cases of non-FIFO behaviour were detected, and there is potential for addressing them in the future through the refinement of the current least-time algorithm. All of these discoveries hold the potential to influence not just sailboat racing but also the utilisation of wind to aid in the propulsion of motor vessels.

In summary, this paper provides comprehensive documentation of the scientific hypotheses and decisions underpinning the development of an open-source ship routing model, while also contributing to the quantification of achievable reductions in greenhouse gas emissions through voyage optimisation.

*Code availability.* Source code of VISIR-2 is available from Salinas et al. (2024a). The distribution includes a user manual.

*Data availability.* Raw data: input datasets and graphs used for the route computations are available from Salinas et al. (2024b); Intermediate data products: routes for producing figures and tables in Sect. 5 are available from Salinas et al. (2024c).

*Video supplement.* Videos for this manuscript are available at https://doi.org/10.5446/s_1687 (ferry case study) and https://doi.org/10.5446/s_1688 (sailboat).

## Appendix A: Angle of attack

The angle $\delta$ between ship's heading and course is obtained from the transcendental equation Eq. 13. Its solution can be approximated by the iteration:

$$\delta^{(0)} = 0 \tag{A1}$$
$$\delta^{(k)} = h(\delta^{(k-1)}) \quad \text{for} \quad k = 1, 2, ...$$

where $k$ is the number of iterations of the function

$$h(x) = \arcsin\left(\frac{\omega_\perp(\delta = x, \delta_i = x - \gamma)}{F(|x - \gamma|)}\right) \tag{A2}$$

with $\gamma$ being a constant resulting from the use of Eq. 2. The $k = 1$ case correspond to the solution provided in Mannarini and Carelli (2019b).

Both Eq. 13 and Eq. A1 were evaluated for a sailboat as in Sect. 5.2.2 using environmental conditions (wind, currents) for a domain in the central Adriatic Sea. 11 hourly time steps and 18,474 edges from a graph with $(\nu, 1/\Delta x) = (4, 12/°)$ were considered, resulting in a total of about $2 \cdot 10^5$ edge weight values. The iterative solution from Eq. A1 was compared to the

880 roots of Eq. 13 found via the `scipy.optimize.root` solver, using as an initial guess the $\delta^{(1)}$ solution of Eq. A1 (see `velocity_eval.py` function in the VISIR-2 code). In what follows, the numerical solution from the solver is termed as "exact". The benefit of the approximated solution Eq. A1 is that it can easily be parallelised on all graph arcs, while this is not possible for the exact solution which processes one arc at time.

The outcome for a sailboat is provided in Fig. A1.a. It is seen that the iterative approximation departs from the exact solution

for $\delta$ angles larger than about 5°. Such departures are mainly related to the effective cross-flow $\omega_\perp$ (marker colour, determining the elongation from the origin). However, it is just a tiny fraction of the edges presenting such departures, so that the $R^2$ correlation coefficient between the exact solution and its approximation is almost identical to 1 for any $k > 0$, as shown in Fig. A1.b.

The case $k = 0$ corresponds to neglecting the loss of ships' momentum to balance the effective cross flow of Eq. 11b.

Therefore, it wrongly underestimates the sailing times. For the First-367 sailboat under consideration here, the $k = 1$ solution leads to a slope about 5% off. Already for $k = 2$ the correct slope is achieved within an error of 2‰. For the ferry, the $k = 1$ iteration is sufficient to reach a 2% accuracy, see Sect. S6.1 in Supplement . This could be due to the ferry having a smoother angular dependence than the sailboat's one, as seen from Fig. 6.b and Fig. 7.a. Finally, in Fig. S22 of the Supplement , evidence of the validation of the exact solution in the absence of currents, Eq. 14, is also provided.

## Appendix B:  Neural network features

For identifying the vessel performance curves from a LUT via a neural network, a multi-layer perceptron was used.

The models were built and trained via the `scikit-learn` package[¶¶¶]. A three-fold cross-validation was used to identify the best model for each vessel performance function. Different solvers, hidden layers' sizes, L2 regularisation terms, and activation functions were explored, covering a search space of about $10^3$ models. The optimal configuration made use of the

900 rectified linear unit activation function, the Adam optimiser to minimise mean-squared error, for at most $10^3$ passes through the training set ("epochs") with a batch size of 200, a constant learning rate of $10^{-4}$ and early stopping after the validation loss has failed to decrease for 10 epochs.

[¶¶¶]https://scikit-learn.org/stable/

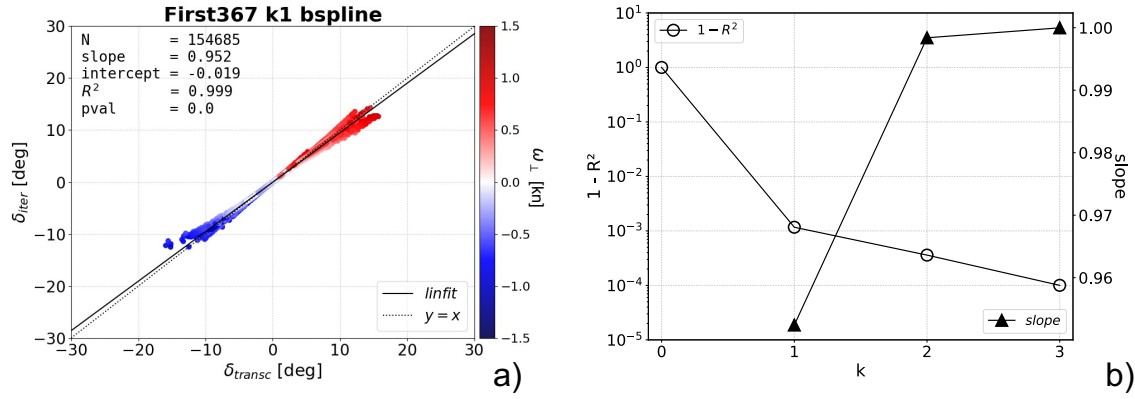

**Figure A1.** Approximate vs. exact solution of Eq. 13 for a First-367 sailboat. a) Iterative solution of Eq. A1 with $k = 1$ vs. the exact solution, using $\omega_\perp$ as marker colour; b) unexplained variance ($R$ is the Pearson's correlation coefficient) of the linear regression and fitted slope coefficient for various $k$ values.

*Author contributions.* G.M.: Conceptualization, Funding Acquisition, Methodology, Project administration, Supervision, Validation, Writing – original draft, Writing – review & editing; M.L.S.: Data Curation, Investigation, Software, Validation, Visualization; L.C.: Data Curation, Investigation, Software, Validation, Visualization; N.P.: Investigation, Resources; J.O.: Investigation, Resources

*Competing interests.* The authors do not declare any competing interests.

*Disclaimer.* The authors are not liable for casualties nor losses occurred in using routes computed via VISIR-2 for navigation purposes.

*Acknowledgements.* Funding through both the Italy-Croatia Interreg projects GUTTA (grant number 10043587) and FRAMESPORT (10253074), as well as the Horizon Europe projects EDITO-ModelLab (101093293) and MISSION (101138583) is acknowledged. Both ChatGPT-3 and Gemini were utilised to review sections of this manuscript for English language accuracy.

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

**Table 1.** Edge count and minimum angle with due North. The total number of edges in the first quadrant is $\nu(\nu+1)$ while the count of non-collinear edges is $N_{q1}$ from Eq. 20. For order of connectivity $\nu$, the angle in the unprojected graph is given by $\Delta\theta = \arcsin(1/\nu)$, regardless of latitude and grid resolution. Using a Mercator projection, at a latitude of $L°$ and for $\Delta x = D°$, the angle is given by $\Delta\theta_D^{(L)}$.

| $\nu$ | $\nu(\nu+1)$ | $N_{q1}$ | $\Delta\theta$ | $\Delta\theta_{1/12}^{(40)}$ | $\Delta\theta_{1/12}^{(70)}$ |
|---|---|---|---|---|---|
| | | | | [°] | |
| 1 | 2 | 2 | 45 | 37.4 | 18.8 |
| 2 | 6 | 4 | 63.4 | 56.9 | 34.3 |
| 3 | 12 | 8 | 71.6 | 66.5 | 45.7 |
| 4 | 20 | 12 | 76 | 71.9 | 53.8 |
| 5 | 30 | 20 | 78.7 | 75.4 | 59.6 |
| 6 | 42 | 24 | 80.5 | 77.7 | 64.0 |
| 7 | 56 | 36 | 81.9 | 79.4 | 67.3 |
| 8 | 72 | 44 | 82.9 | 80.7 | 69.9 |
| 9 | 90 | 56 | 83.7 | 81.7 | 72.0 |
| 10 | 110 | 64 | 84.3 | 82.6 | 73.7 |

**Table 2.** Principal parameters of the ferry.

| Name | Symbol | Value | Units |
|---|---|---|---|
| Length overall | LOA | 125 | m |
| Draft middle | $T$ | 5.3 | m |
| Deadweight | DWT | 4,050 | t |
| Main engine power | $P_{\mathrm{main}}$ | 4,000 | kW |
| Main engine rated speed | $n_{\mathrm{eng}}$ | 750 | rpm |
| Service speed | $v_S$ | 19 | kn |

**Table 3.** Principal parameters of the sailboat (First-367).

| Name | Symbol | Value | Units |
|---|---|---|---|
| Length of hull | $L_{\mathrm{hull}}$ | 10.7 | m |
| Draft | $T$ | 2.2 | m |
| Displacement volume | $\nabla$ | 5.8 | m$^3$ |
| Rudder wetted surface | - | 1.4 | m$^2$ |
| Keel wetted surface | - | 3.3 | m$^2$ |
| Main sail area | - | 38 | m$^2$ |
| Jib sail area | - | 4 | m$^2$ |
| Spinnaker area | - | 95 | m$^2$ |

**Table 4.** VISIR-2 modules with their original names, purpose, and references within this paper. Modules #1-5 represent the core package and require the visir-venv virtual environment. Module #6 runs with visir_vis-venv.

| # | Module name | Meaning | Reference |
|---|---|---|---|
| 1 | Grafi | graph geometry | Sect. 2.2 |
| 2 | Campi | environmental fields | Sect. 2.3 |
| 3 | Navi | vessel performance | Sect. 2.5 |
| 4 | Pesi | edge weights | Sect. 2.3 |
| 5 | Tracce | shortest path | Sect. 2.4 |
| 6 | Visualizzazioni | rendering | Sect. 2.6 |
| 7 | GUI | graphical user interface | - |
| 8 | PostProc | assessments of products | - |
| 9 | Validazioni | benchmark runs | - |
| 10 | Utilità | various utilities | - |
| 11 | Docs | documentation | - |

**Table 5.** VISIR-2 route durations compared to analytic oracles (Cycloid and Techy), as referenced in the main text. $L_0$ and $T_0$ represent the length and time scales, respectively. The oracle durations are denoted by $T^{(e)}$, while those from VISIR-2 are denoted by $T^*$. The percentage mismatch is calculated as $dT^* = (T^*/T^{(e)}) - 1$.

| oracle | $\nu$ | $1/\Delta x$ $1/^{\circ}$ | $\Delta\tau$ min | $L_0$ nmi | $T_0$ hr | $T^{(e)}$ $T_0$ | $T^*$ $T_0$ | $dT^*$ % |
|---|---|---|---|---|---|---|---|---|
| | 2 | 60 | 5 | | | | 1.738 | 0.691 |
| Cycloid | 5 | 60 | 5 | 56.38 | 2.672 | 1.726 | 1.726 | 0.012 |
| | 10 | 50 | 5 | | | | 1.732 | 0.342 |
| | 5 | 25 | 5 | | | | 1.057 | 0.076 |
| Techy | 5 | 100 | 5 | 140.11 | 6.640 | 1.056 | 1.046 | -0.956 |
| | 10 | 50 | 5 | | | | 1.050 | -0.599 |

**Table 6.** VISIR-2 vs. LSE durations. The relative error $dT_{\mathrm{res}}^*$ is defined as the discrepancy between $T^*$ and LSE at two different grid resolutions $1/\Delta x$. Both VISIR-2 and VISIR-1 outcomes are provided.

| model | $\nu$ | $1/\Delta x$ $1/°$ | $\Delta\tau$ min | $T^*$ hr | $dT_{120}^*$ % | $dT_{240}^*$ % |
|---|---|---|---|---|---|---|
| VISIR-1.b | 6 | 129 | - | 13.73 | -1.58 | -0.43 |
| VISIR-2 | 6 | 129 | 30 | 13.62 | -2.36 | -1.23 |
| VISIR-1.b | 3 | 134 | - | 13.79 | -1.12 | 0.03 |
| VISIR-2 | 3 | 134 | 30 | 13.71 | -1.73 | -0.59 |
| VISIR-1.b | 2 | 142 | - | 13.90 | -0.36 | 0.79 |
| VISIR-2 | 2 | 142 | 30 | 13.85 | -0.74 | 0.42 |
| LSE | - | 120 | - | 13.95 | - | - |
| | - | 240 | - | 13.79 | - | - |

**Table 7.** VISIR-2 vs. openCPN comparison. Durations $T^*$ and relative mismatch $dT^*$ for the cases shown in Fig. 9 are provided. $k = 2$ and $\Delta\tau = 15$min used throughout the numerical experiments.

| version | $\nu$ | $1/\Delta x$ [1/deg] | wind | | | | current + wind | | | |
|---|---|---|---|---|---|---|---|---|---|---|
| | | | Westbound | | Eastbound | | Westbound | | Eastbound | |
| | | | $T^*$ [hr] | $dT^*$ [%] | $T^*$ [hr] | $dT^*$ [%] | $T^*$ [hr] | $dT^*$ [%] | $T^*$ [hr] | $dT^*$ [%] |
| VISIR-2 | 4 | 12 | 51.9 | 3.4 | 34.4 | -0.4 | 54.0 | -2.6 | 32.2 | 0.0 |
| | 5 | 15 | 52.0 | 3.5 | 34.5 | -0.2 | 53.9 | -2.7 | 31.7 | -1.5 |
| | 6 | 18 | 51.2 | 1.9 | 33.6 | -2.9 | 53.4 | -3.7 | 30.9 | -3.9 |
| | 7 | 21 | 50.7 | 1.0 | 32.8 | -5.0 | 52.8 | -4.8 | 30.9 | -4.1 |
| | 8 | 23 | 51.0 | 1.6 | 32.8 | -5.0 | 53.1 | -4.2 | 30.8 | -4.5 |
| openCPN | | | 50.2 | | 34.6 | | 55.4 | | 32.2 | |

| submodule | $a$ [us] | $p(a)$ | $b$ [us] | $p(b)$ | $c$ [s] | rmse [s] |
|---|---|---|---|---|---|---|
| dist_D | 0.030 | 0.001 | 1.120 | 0.000 | | 0.245 |
| dist_tot | 0.001 | 0.005 | 1.333 | 0.000 | | 0.638 |
| time_D | 1.814 | 0.000 | 1.003 | 0.000 | | 0.328 |
| time_tot | 1.111 | 0.000 | 1.031 | 0.000 | | 0.161 |
| CO2t_D | 0.952 | 0.000 | 1.037 | 0.000 | | 0.191 |
| CO2t_tot | 0.594 | 0.000 | 1.064 | 0.000 | | 0.646 |
| time_D_V1b | 26.000 | | 1.010 | | 0 | |
| time_tot_V1b | 1.200 | | 1.180 | | 60 | |

**Table 8.** Fit coefficients of the $T_c = a \cdot DOF^b + c$ regressions for various components of `Tracce`, motor vessel version. "D" stands for the Dijkstra's algorithm only, while "tot" includes the post-processing for reconstructing the voyage. $p(K)$ is the $p$-value for the $K$ coefficient. All data refers to VISIR-2 but the ∗_V1b ones, referring to VISIR-1.b.

| | ITPTO - FRTLN | | | | | FRTLN - ITPTO | | | | |
|---|---|---|---|---|---|---|---|---|---|---|
| | $\chi$ [%] | | | | | $\chi$ [%] | | | | |
| | 70 | 80 | 90 | 100 | avg | 70 | 80 | 90 | 100 | avg |
| wa | 2.9 | 2.2 | 1.4 | 1 | 1.9 | 0.9 | 0.6 | 0.4 | 0.3 | 0.6 |
| wa-cu | 3.4 | 2.5 | 1.7 | 1.2 | 2.2 | 1.4 | 1.2 | 0.7 | 0.6 | 1 |

**Table 9.** Average relative savings of the $CO_2$ -optimal vs. the least-distance route (in %), for various engine loads ($\chi$), considering just waves (wa) or also currents (wa-cu), for ferry routes between Toulon (FRTLN) and Porto Torres (ITPTO) as in Fig. 11. The $\chi$-averaged values are also provided in the "avg" columns.

| $\chi$ | $a$ | $b$ | $d_1$ | $d_2$ |
|---|---|---|---|---|
| [%] | [-] | [-] | [%] | [%] |
| 70 | 638 | 0.017 | 1.7 | 29.1 |
| 80 | 646 | 0.022 | 1.6 | 19.0 |
| 90 | 662 | 0.030 | 1.3 | 11.3 |
| 100 | 667 | 0.029 | 1.4 | 8.8 |

**Table 10.** Fit coefficients of $y = a \cdot [\exp(-x/d_1) + b \cdot \exp(-x/d_2)]$ on the data of Fig. 12.b.

|          | GRMON - TRMRM | | | TRMRM - GRMON | | |
|----------|--------|-------------|-------------|--------|-------------|-------------|
|          | $-dT^*$ | $N_f^{(g)}$ | $N_f^{(o)}$ | $-dT^*$ | $N_f^{(g)}$ | $N_f^{(o)}$ |
| wi       | 2.5 | 256 | 1 | 2.3 | 308 | 1 |
| wi-le    | 2.3 | 275 | 1 | 2.4 | 328 | 3 |
| wi-cu    | 3.1 | 254 | 1 | 3.1 | 320 | 1 |
| wi-cu-le | 3.2 | 267 | 0 | 3.5 | 326 | 6 |

**Table 11.** Average time savings of the sailboat routes (in %), considering just wind (wi), or also various combinations of currents (cu) and leeway (le), for the sailboat routes between Monemvasia (GRMON) and Marmaris (TRMRM) as in Fig. 13. The number of failed routes for the least-distance $N_f^{(g)}$ or the least-time routes $N_f^{(o)}$ is also provided.