# Peer review of "VISIR-2: ship weather routing in Python"

_EGUsphere, 2023_

## Author Comment (AC1)

**RC1**: 'Comment on egusphere-2023-2060', Anonymous Referee #1, 30 Nov 2023

**AC1:** Thank you for dedicating your time to the assessment of our manuscript. Below, the Referee's text is presented in black, and our response in blue; proposed changes to the manuscript are typed in red. Please be aware that the figure, table, and section numbering pertain to the revised version of the manuscript. Additional references are provided at the end of this document.

I recommend that this manuscript is accepted for publication with minor revisions.
The submitted manuscript describes new updates to the VISIR-2 weather routing software. The model has been developed from Matlab into Python and is available as an open-source model with no license.
A brief literature review is provided, alongside an in-depth methodology, providing a detailed explanation of each component of the model. The paper finishes by demonstrating VISIR-2's ability to minimise CO2 emissions in two case studies – one for a motor ship and one for a sailing ship.
The paper excels in its scientific reproducibility and I would like the congratulate the authors on the well-organised manuscript with impressive levels of additional detail provided, including the user manual. I believe that the modular implementation, released under a freely available GNU General Public License, has strong uses for the scientific community and the wider community. The paper excels in its presentation quality, in particular the presentation of the very clear figures. VISIR-2 benefits greatly from its validation and provides very useful and detailed insight into the model's computation time.
We sincerely appreciate the reviewer for dedicating their time and providing valuable feedback on our manuscript. Their insightful comments were instrumental in enhancing the quality and the presentation of our work.

That said, I believe the paper could improve in the following aspects.
I believe the manuscript would benefit from a clearer description of the inherent novelty of the model within the method section. The novel aspects of the model are not highlighted sufficiently in the method, and I finished reading the section wondering which parts of VISIR-2 form the novel features. I would also like to see this same novelty more clearly described in the introduction – here, the authors describe additional features well, but the section fails to comment on the novelty of these features in the context of the body of literature in this field.
The paper heavily focuses its novelty on the fact that their model has been developed as open source, modular and free to use (which VISIR-2 largely does with exceptional care and quality). While this is indeed useful, I believe the manuscript could benefit from a discussion on the novel quantitative implications of their CO2-saving results. (In fact, the authors also touch upon this in the first paragraph of their abstract: "…its quantitative impact has been explored only to a limited extent…".)
Thank you for emphasizing the need to better elucidate the novelty of VISIR-2 across the various sections of our manuscript. Indeed, its novelty extends beyond being an open-source model. VISIR-2 introduces several technical innovations and we are going to highlight the absolute novelties at various points throughout the revised manuscript. To summarise them:
- the unified modelling framework to account for involuntary ship speed loss through waves (Sect.2.1) now mentioned in Sect.1

- the capacity to account for currents advection for a vessel with a angle-dependent performance curve, and the solution of corresponding transcendental equation (Sect.2.1 and App.A) now mentioned in Sect.2.1
- the use of a previously overlooked cartographic projection (Sect.2.2.1) now mentioned in Sect.2.2.1
- the use of a K-dimensional tree for indexing graph edges (Sect.2.2) now mentioned in Sect.2.2.3
- various options for spatio-temporal interpolation of environmental fields (Sect.2.3) mentioned in Sect.2.3.2
- a novel and versatile optimisation algorithm in presence of dynamic edge weights (Sect.2.4), here applied to minimize $CO_2$ emissions but adaptable to other metrics as well, like passenger comfort or radiated underwater noise now mentioned in Sect.2.4
- the visualization of the dynamic environmental fields along the route making use of isochrone-bounded sectors (Sect.2.6) now mentioned in Sect.2.6
- the modular structure of the software suite, benefitting R&D activities (Sect.2.7) mentioned in Sect.2.7
- the investigation of the distribution of the $CO_2$ savings, revealing a bi-exponential pattern Sect.5.2.1) now mentioned in Sect.5.2.1
- the use of wind, currents, and leeway in the computation of sailboat optimal routes (Sect.5.2.2) now mentioned in Sect.5.2.2

I don't currently fully understand what novel academic question they try to answer with their analysis, or whether it is just used to showcase VISIR-2 (which it does a great job of). I believe that these quantitative novelties are present in the paper, but more work is needed to outline them in the results/discussion section. While outside the area of model development, the results of this paper are great work and I believe it would be of use to advance the field more generally. I would ideally like to see an explanation of how their quantitative results/discussion contribute to new science. This could also be brought out briefly in the abstract.

We contend that the quantification of ship weather routing's impact on reducing greenhouse gas (GHG) emissions still represents a significant gap in the current scientific literature. A wide range of percentage savings are reported, often lacking sufficient information to precisely understand the methodologies employed to achieve them. The situation becomes even more challenging when seeking open-source models. Indeed, we only encountered the openCPN system. As the maritime industry endeavors to adhere to the resolutions set forth by the International Maritime Organization (IMO) for emission reduction, it is imperative to meticulously evaluate the precise potential of both technical and operational measures to achieve this objective within the stipulated time frame. This is undoubtedly a subject area where we believe a thoroughly documented, open-source model like VISIR-2 could be highly relevant. Other potential applications are also conceivable, including the establishment of baselines for emission profiles. Such capabilities could be particularly valuable for agencies like the European Maritime Safety Agency (EMSA), which implements a system for monitoring, reporting, and verifying vessels' $CO_2$ emissions. Starting in 2024, maritime GHG emissions are incorporated into the European Union Emissions Trading System (EU-ETS). This entails shipowners, ship managers, or bareboat charterers (whoever bears the cost of fuel) surrendering allowances for their emissions within the EU-ETS. Saving fuel, such as through the implementation of smarter routes, has become

increasingly essential in light of these developments. Even with the adoption of zero- or low-carbon fuels like e-methanol or green ammonia, it remains critical to conserve these fuels as much as possible. This is particularly important considering that their unit price is significantly higher than traditional fossil fuels.

We are going to allocate the provided text concerning motivations for VISIR-2 across the Abstract, Introduction (Sect. 1), the Discussion (Sect. 6.2), and the Conclusions (Sect. 7) of the revised manuscript. In particular, regarding the quantitative contribution of the VISIR-2 model to new science, we have introduced a new Discussion section. It incorporates the aforementioned motivations for VISIR-2 and includes a detailed quantitative comparison with existing literature, as outlined in our response to the Referee's subsequent comment.

On a similar note, the results/discussion does a great job of highlighting the potential of weather routing as a CO2 reduction measure. However, there is almost no discussion on how their results compare to other studies in the literature. Where do their CO2 savings fit in the literature? Do they agree/disagree? What relative contribution is the paper making to this body of literature? I would like to see more discussion on this.

When comparing the carbon savings of VISIR-2 with those reported in the literature on ship weather routing, it is notable that only a limited number of peer-reviewed papers have addressed emission savings through ship routing thus far.  A few findings available for comparison with the results of VISIR-2 are presented in what follows.

For the ferry case study examined in this manuscript, the $CO_2$ emissions, in the best-case scenario, can be halved compared to those along the least-distance route (Fig. 12). This is in numerical agreement with the broad (0.1 - 48)% range reported in Bouman et al. (2017)[Tab.2]. Notably, the upper limit reduction stands as an outlier, with the more probable values depicted in Bouman et al. (2017)[Fig.2] falling below 10%. This resonates well with the outcome presented in Fig. 12.b and Tab. 9 of the revised manuscript, based on thousands of reproducible numerical simulations for a ferry obtained via VISIR-2.

Applying the VOIDS model, Mason et al. (2023a)[Sect.3.2] discovered that, for eastbound routes of a Panamax bulk carrier in the North Atlantic, voyage optimisation contributed to carbon savings ranging from 2.2 to 13.2%. This is a narrower range compared to the present findings of VISIR-2. However, both a different vessel type (a ferry) and a different domain of sailing (the Western Mediterranean Sea) were considered in the VISIR-2 numerical experiments.

Miola et al. (2011) presented data from the second IMO GHG study, where the estimated CO2 abatement potential for weather routing on the emissions of 2020 was reported to be as low as 0.24%. In the same paper, also a DNV study on projected emissions in 2030 is cited, providing an estimate of 3.9% for the CO2 abatement potential through weather routing. The former figure compares well to the average emission reduction computed via VISIR-2 for the ferry downwind conditions and high engine load, the latter to results for upwind and low engine load (cf. Tab.8).

Lindstad et al. (2013) estimated the reduction in $CO_2$ emissions for a dry bulk Panamax vessel navigating in head seas during a typical stormy period in the North Atlantic. This reduction was determined when sailing on a 4,500 nautical miles (nmi) route compared to

the shorter (yet stormier) least-distance route of 3,600 nmi. They found reductions ranging from 11 to 48%, depending on the speed of the vessel.

We note that, as e.g. in Mason et al.(2023a)[Fig.7], also VISIR-2 optimal routes exhibit spatial variations contingent on the departure date, forming a "bundle" as illustrated in Fig. 11.b. The shape of route boundaries was assessed for the United States' Atlantic coast routes by means of AIS data in Breithaupt et al. (2017). While they found multimodal distributions depending on sailing direction, they did not attribute the preferential lanes to the presence of ocean currents but speculated that it was due to bathymetric constraints or artificial aids to navigation.

VISIR possesses a capability to incorporate ocean currents into the voyage optimisation process. As shown in Mannarini and Carelli (2019), this integration has proven to significantly reduce the duration of transatlantic routes. In the present manuscript, we reaffirm the positive impact of currents on ship route optimisation, extending their benefits also to the reduction of $CO_2$ emissions (Tab. 8) and to the determination of more faithful duration savings for sailboat routes 55 (Tab. 10).

In general, both average and extreme $CO_2$ emission percentage savings found in literature align well with the results obtained in the ferry case study presented in our manuscript. Nevertheless, engaging in a meaningful discussion of numerical differences, given the diverse range of vessel types, routes, environmental fields, and computational methods employed in the various published case studies, proves challenging.

VISIR-2 contributes to the existing body of literature by providing an open computational platform that facilitates the simulation of optimal ship routes in presence of waves, currents, and wind. These simulations are designed to be transparent, with customisable sea domain and vessel performance curves, allowing for thorough inspection, modification, and evaluation. This addresses the concern raised by Zis et al. (2020) regarding the necessity of benchmarking instances of optimal routes and the associated input data. By providing such benchmarks, VISIR-2 supports and streamlines the work of future researchers in the field. Hence, we believe that the critical task of evaluating inter- model differences will best be addressed through dedicated inter-comparison studies. As previously demonstrated with VISIR-1 (Mannarini et al., 2019), similar assessments could be conducted with VISIR-2 too. The above text is going to be added in Sect.6.1.

That said, the level of detail in the results/discussion is brilliant and very commendable, a great job on that.
Some of my above suggestions have been completed to some extent in the conclusion - but should be strengthened in the other sections. No new information should be presented in the conclusion.
Thanks for this feedback.
As noted earlier, we have incorporated a new "Discussion" section (Sect.6) into the revised manuscript to specifically address this recommendation by the Referee. Additionally, the Conclusions have been revised to emphasize the results and their novelty while refraining from introducing any new information for the first time in the manuscript.

Alongside this, I have the following minor comments:

1. I believe a figure at the start of the method section that provides an outline of all steps of the model would help readers to understand the model structure more generally

   We are going to add a figure in Sect.2.7 providing a depiction of the VISIR-2 workflow across the various code modules:

[Figure]

   **Figure 8.** VISIR-2 workflow. Modules enclosed within thicker frames are intended for execution by the end-user, while the other modules can be modified for advanced usage. The data flow occurs along the wavy arrow, with routine calls along the dashed line.

2. h-hat is mentioned at the end of page 5 but I cannot see this variable in Figure 1.

   Thank you for noting this.

   We fix it by replacing the "$s$" versor and subscript with "$h$" in both the main text and Fig. 1.

3. Removing collinear edges is a great idea, one which I will test myself!

   Thank you for your feedback. Please be aware that we are renaming "collinear" to "quasi-collinear edges" in the revised manuscript, as their directions exhibit slight differences once a cartographic projection is applied. This will be detailed in a new Sect.2.2.3.

4. I found Section 2.3.2 Space interpolation confusing. I would consider rewording. A better description would be useful. Same with Figure 5. Please explain the meaning of Head and Tail and give a more clear description of the two interpolation methods. Indeed, please state the difference and why the two are necessary.

   Each graph edge acts as an arrow, conveying flow from ``tail'' to ``head'' nodes.

   This is being clarified in Sect.2.2.2. and Fig.5a. Fig.5b makes clear the different impact of the two interpolation schemes as shown below.

[Figure]

   **Figure 5.** Spatial interpolation in VISIR-2. a) The squares represent grid nodes of the environmental field $\varphi(x)$, and the filled circles graph grid nodes. A graph edge is depicted as a magenta segment. b) Transect of a) along the edge direction $\hat{e}$, with the interpolator of $\varphi$ as a grey solid line. The $(0,1)$ subscripts refer to the value of the Sint parameter while $h$ and $t$ to the edge head and tail, respectively.

We have deeper investigated the effect of the two interpolation schemes, Sint=0 and Sint=1 which were originally devised in an attempt to reproduce benchmark results from VISIR-1 obtained from the *interp2* Matlab function.

We generated ideal z = f(x,y) fields: a plane, a paraboloid, and a family of saddles. Also, we generated a set of graph edges in the (x,y) plane, with different lengths and orientations. We interpolated the fields on each edge, using both schemes. We found that, as expected, as the edge length is reduced, the results for Sint=0 and Sint=1 converge to a common truth. However, depending on the specific saddle hypersurface and edge orientation, either Sint=0 or Sint=1 converge quicker.

Also, upon closer examination, the computational performance was found to be contrary to what was previously stated in the preprint: the Sint=1 option (evaluating the environmental field at the edge barycenter) is not computationally faster than the Sint=0 option (average of field values at the head and tail). Actually, the opposite holds true. The reason being that, in the case of Sint=1, the interpolator is applied at each edge, whereas with Sint=0, it is applied at each node. Given that the number of edges exceeds the number of nodes by a factor defined by Eq.20, for the degree of connectivity of the case studies ($v$ parameter) the computational time for Sint=1 is found to be approximately one order of magnitude higher.

Therefore, the new default scheme is set to be Sint=0.

The above text is used for the new Sect. 2.3.2. It refers to the new Supplement's figure which is reported also below.

[Figure]

**Fig. S0** a)-c) Test hypersurfaces (shaded in grey) and graph edges (coloured lines and markers indicating the edge head). d)-f) Edge representative values of the different hypersurfaces for both Sint=0 (circles) and Sint=1 (triangles).

In the revised VISIR-2 source code, we have now designated Sint=0 as the default interpolation scheme.

5. The least squares fit for the blue lines in Figure 9 doesn't seem to work. I'm not sure if an alternative is possible, but if my interpretation is correct, the solid blue line (total computation time) should not fall below the dashed blue line (Dijkstra component of computation time). While your interpretation is correct, we are still confident in the goodness of the fit. Minimal nonlinearities exist in the performance of both the total and Dijkstra's component of the least-distance routine, particularly

for numerical problems with a small number of degrees of freedom. However, the mismatch between the data and the fit line is minimal, typically a matter of a few tenths of a second (note the log-log scale in Fig.10). Notably, the power-law function successfully fits the computing time above a few tens of seconds, corresponding to numerical problems of more realistic size.

In fact, this is the same for the red and green lines also.

The red and green lines represent the least-time and least-CO2 routines, respectively. Due to their computation times being one order of magnitude larger than those of the least-distance routine, they are less sensitive to the nonlinearity.

All fit coefficients are found to be highly statistically significant and, for all fit models, the root mean square error (rmse) is confirmed to be smaller than one second.

To showcase the goodness of fit, we have incorporated both the rmse and the fit coefficients' *p*-values into Tab.8, making a corresponding point in the text.

| submodule | $a$ [us] | $p(a)$ | $b$ [us] | $p(b)$ | $c$ [s] | rmse [s] |
|---|---|---|---|---|---|---|
| dist_D | 0.030 | 0.001 | 1.120 | 0.000 | | 0.245 |
| dist_tot | 0.001 | 0.005 | 1.333 | 0.000 | | 0.638 |
| time_D | 1.814 | 0.000 | 1.003 | 0.000 | | 0.328 |
| time_tot | 1.111 | 0.000 | 1.031 | 0.000 | | 0.161 |
| CO2t_D | 0.952 | 0.000 | 1.037 | 0.000 | | 0.191 |
| CO2t_tot | 0.594 | 0.000 | 1.064 | 0.000 | | 0.646 |
| time_D_V1b | 26.000 | | 1.010 | | 0 | |
| time_tot_V1b | 1.200 | | 1.180 | | 60 | |

**Table 8.** Fit coefficients of the $T_c = a \cdot DOF^b + c$ regressions for various components of Tracce, motorboat version. "D" stands for the Dijkstra's algorithm only, while "tot" includes the post-processing for reconstructing the voyage. $p(K)$ is the $p$-value for the $K$ coefficient. All data refers to VISIR-2 but the *_V1b ones, referring to VISIR-1.b.

6. Section 5.1 Environmental fields – should the URL next to "a lower resolution (0.4, URL)" go as a footnote?
   Done.
   To be changed in Sect.5.1

7. Figure 10 b – super clear and very interesting. An engaging plot.
   Thank you for this feedback.

**References**

[Bouman_2017] Evert A. Bouman, Elizabeth Lindstad, Agathe I. Rialland, Anders H. Strømman, State-of-the-art technologies, measures, and potential for reducing GHG emissions from shipping – A review, Transportation Research Part D: Transport and Environment, Volume 52, Part A, 2017, Pages 408-421, https://doi.org/10.1016/j.trd.2017.03.022

[Breithaupt_2017] Breithaupt, S.A., Copping, A., Tagestad, J. and Whiting, J., 2017. Maritime route delineation using AIS data from the Atlantic coast of the US. The Journal of Navigation, 70(2), pp.379-394.

[Lindstad_2013b] Haakon Lindstad, Bjørn E. Asbjørnslett, Egil Jullumstrø, Assessment of profit, cost and emissions by varying speed as a function of sea conditions and freight market, Transportation Research Part D: Transport and Environment, Volume 19, 2013, Pages 5-12, https://doi.org/10.1016/j.trd.2012.11.001

[Mannarini_2019] Mannarini, G., Subramani, D., Lermusiaux, P., and Pinardi, N.: Graph-Search and Differential Equations for Time-Optimal Vessel Route Planning in Dynamic Ocean Waves, IEEE Transactions on Intelligent Transportation Systems, 21, 3581–3593, https://doi.org/10.1109/TITS.2019.2935614, 2019.

[Mannarini_2019b] Mannarini, G. and Carelli, L.: VISIR-1.b: ocean surface gravity waves and currents for energy-efficient navigation, Geoscientific Model Development, 12, 3449–3480, https://doi.org/https://doi.org/10.5194/gmd-12-3449-2019, 2019.

[Mason_2023b] James Mason, Alice Larkin, Simon Bullock, Nico van der Kolk, John F. Broderick, Quantifying voyage optimisation with wind propulsion for short-term CO2 mitigation in shipping, Ocean Engineering, Volume 289, Part 1, 2023, 116065, https://doi.org/10.1016/j.oceaneng.2023.116065

[Miola_2011] A. Miola, M. Marra, B. Ciuffo, Designing a climate change policy for the international maritime transport sector: Market-based measures and technological options for global and regional policy actions, Energy Policy, Volume 39, Issue 9, 2011, Pages 5490-5498, https://doi.org/10.1016/j.enpol.2011.05.013

[Zis_2020] Thalis P.V. Zis, Harilaos N. Psaraftis, Li Ding, Ship weather routing: A taxonomy and survey, Ocean Engineering, Volume 213, 2020, https://doi.org/10.1016/j.oceaneng.2020.107697

---

## Author Comment (AC2)

**RC2**: 'Comment on egusphere-2023-2060', Anonymous Referee #2, 09 Jan 2024
**AC2:** Thank you for dedicating your time to the assessment of our manuscript. Below, the Referee's text is presented in black, and our response in blue; proposed changes to the manuscript are typed in red. Please be aware that the figure, table, and section numbering pertain to the revised version of the manuscript. Additional references are provided at the end of this document.

**General Comments**

Overall the manuscript is of high quality and provides a very thorough analysis on a proposed navigational approach for both motorboats and environmentally-driven/impacted surface vessels. Currents and leeways are compared for two separate boat models and meticulously analyzed seasonally over a scoped domain, offering high quality conclusions and results discussion, while also offering rich model bases for the open literature. Another open-source weather routing software is invaluable as prior to this, primarily only openCPN was the go-to open-source option that would not be able to handle motorboat and CO2-based measures of optimization. The primary contribution is its attention to detail and reproducibility for science computation that is wholly lacking on the open-source playing field.

We extend our gratitude to the reviewer for investing their time and delivering precise feedbacks on our manuscript. Their insightful observations have significantly contributed to both debugging the model code and improving its presentation in this manuscript.

**Specific Comments**

There are still a handful of revisions I believe the paper needs to undergo to be finalized for publication.

Throughout the manuscript, there are numerous undefined acronyms in this section that either need to be noted as a footnote, or explained to the reader. E.g., GFS, OSCAR, AVALON, GUTTA, openCPN, especially in Sec 1.1.1.

Whenever feasible, we have addressed this issue. Nevertheless, an acronym expansion for the AVALON service remains undetermined.

Acronyms expanded in Sect.1.1.1:

GFS - Global Forecast System

GUTTA - savinG fUel and emissions from mariTime Transport in the Adriatic region

NOAA - National Oceanic and Atmospheric Administration

openCPN - open-source Chart Plotter Navigation

OSCAR - Ocean Surface Current Analyses Real-time

VISIR - discoVerIng Safe and effIcient Routes

The scientific notation is very hard to follow. It is very hard to distinguish a vector quantity from a scalar. Can you use typographic convention to aid the reader? Hard-to-follow notational conventions induces increased effort in assessing the research contributions in Sec. 2.1 because of this mismatch and lack of clarity.

Thank you for the suggestion. The bold font previously utilised for representing vector quantities has now been substituted with a vector arrow. New vector quantities, such as $\overrightarrow{G}$ = SOG $\hat{e}$, have been introduced.
To be changed in Sect.2.1.

In addition, the authors flip between radians and degrees. For consistency and legibility, they should remain the same throughout unless where deemed necessary for more intuitive understanding for the reader.
Thank you for bringing this to our attention. We have now adopted the use of degrees consistently throughout the manuscript and the model's source code.
To be changed in Sect.2.1.

Sec 5.1: Should more than just surface current be used for the ferry? It seems the draft is in excess of 4m, so potentially 0, 2, and 4m relative z-levels could be employed for even further increased fidelity in the optimization at the expense of computational complexity.
The kinematics of VISIR-2 presented in Sect.2.1 do not inherently limit the use to just surface ocean currents. This was just an initial approximation based on the literature discussed in [Mannarini_2019]. However, multi-sensor observations reported in [Laxague_2018] at a specific location in the Gulf of Mexico revealed a significant vertical shear, both in magnitude (by a factor of 2) and direction (by about 90 degrees), within the first 8 metres. Numerical ocean models typically resolve this layer, for instance the Mediterranean product of CMEMS provides four levels within that depth. This vertically resolved data holds the potential to refine the computation of a ship's advection by the ocean flow. A plausible approach could involve the linear superposition of the vessel vector velocity with a weighted-average of the current, considering also the ship's hull geometry.
We are going to add this text in Sect.5.1 and mention it also in the outlook subsection of the Discussion (Sect.6.3).

There is a large and important question when assessing graph edges throughout the paper and that is what coordinate system/projection/transformation is assumed. This is a very important piece of information missing from a geodesy and nautical navigation standpoint.
Thank you for bringing this to our attention; indeed, your observation is accurate. Upon thorough examination, we identified that we overlooked a cartographic projection in the graph grid of VISIR. So far, and just for the visualisation of the routes, an equirectangular projection or plate-carrée was used.
To fix this issue, we have updated the VISIR-2 model code to ensure that a projection is considered also for the computation of the graph edge directions, for the intersections between edges and shoreline segments, and during the environmental fields processing. We made use of the *pyproj* library for converting the original lat/lon information of the WGS-84 ellipsoid into a Mercator projection. This specific projection was chosen for its conformality and for leading to straight images of constant-bearing lines, a convenient feature for navigational purposes [Feeman_2002]. The reference parallel was taken to be the equator.
In the visualisation module, the *cartopy* library has been introduced and used to render maps in Mercator projection.
Additional details on this important matter can be found in our responses to subsequent Referee's comments below. We here anticipate the finding that the missing projection had a relatively minor impact on edge direction or ship course (less than a 6-degree error) in the

case studies, primarily due to the intermediate latitudinal range utilised, as shown in the table below.

| Case study | Average latitude [°] | Graph grid parameters | | Angle closest to due North [°] | |
|---|---|---|---|---|---|
| | | $\nu$ | $1/\Delta x$ [1/°] | No projection | Mercator |
| ferry | 42 | 4 | 12 | 71.6 | 65.8 |
| sailboat | 36 | 5 | 15 | 76.0 | 72.8 |

However, considering a cartographic projection is particularly relevant for vessels whose performance curve is highly sensitive to the angle of attack of environmental fields, such as sailboats. Indeed, we noted that the projection results in an improvement in the validation outcomes of VISIR-2 compared to openCPN, as shown in Tab.7. The contents of the table are detailed below:

no projection:

| | | | | wind | | | | current + wind | | | |
| | | | | Westbound | | Eastbound | | Westbound | | Eastbound | |
| version | $\nu$ | invDx | $\Delta\Theta$ | $T^*$ | $dT^*$ | $T^*$ | $dT^*$ | $T^*$ | $dT^*$ | $T^*$ | $dT^*$ |
| | | [1/deg] | [deg] | [hr] | [%] | [hr] | [%] | [hr] | [%] | [hr] | [%] |
| VISIR-2 | 4 | 12 | 14 | 55.1 | 9.7 | 34.6 | 0.2 | 57.7 | 4.0 | 32.3 | 0.2 |
| | 5 | 15 | 11 | 54.9 | 9.3 | 34.5 | 0.0 | 57.2 | 3.2 | 31.6 | -1.9 |
| | 6 | 18 | 9 | 54.3 | 8.0 | 33.4 | -3.4 | 56.4 | 1.8 | 31.0 | -3.7 |
| | 7 | 21 | 8 | 53.7 | 6.9 | 32.9 | -4.7 | 55.4 | -0.1 | 30.8 | -4.3 |
| | 8 | 23 | 7 | 54.0 | 7.5 | 32.9 | -4.7 | 56.2 | 1.3 | 30.9 | -4.0 |
| openCPN | | | | 50.2 | | 34.6 | | 55.4 | | 32.2 | |

with projection:

| | | | | wind | | | | current + wind | | | |
| | | | | Westbound | | Eastbound | | Westbound | | Eastbound | |
| version | $\nu$ | invDx | $\Delta\Theta$ | $T^*$ | $dT^*$ | $T^*$ | $dT^*$ | $T^*$ | $dT^*$ | $T^*$ | $dT^*$ |
| | | [1/deg] | [deg] | [hr] | [%] | [hr] | [%] | [hr] | [%] | [hr] | [%] |
| VISIR-2 | 4 | 12 | 14 | 51.9 | 3.4 | 34.4 | -0.4 | 54.0 | -2.6 | 32.2 | 0.0 |
| | 5 | 15 | 11 | 52.0 | 3.5 | 34.5 | -0.2 | 53.9 | -2.7 | 31.7 | -1.5 |
| | 6 | 18 | 9 | 51.2 | 1.9 | 33.6 | -2.9 | 53.4 | -3.7 | 30.9 | -3.9 |
| | 7 | 21 | 8 | 50.7 | 1.0 | 32.8 | -5.0 | 52.8 | -4.8 | 30.9 | -4.1 |
| | 8 | 23 | 7 | 51.0 | 1.6 | 32.8 | -5.0 | 53.1 | -4.2 | 30.8 | -4.5 |
| openCPN | | | | 50.2 | | 34.6 | | 55.4 | | 32.2 | |

The improvement is especially noticeable for the upwind routes ("westbound" in table), where maximum errors decreased from approximately 9 to 3%. (Tab.7, or Tab.6 in the preprint, was also affected by a compilation error). Indeed, even a slight deviation in course could result in wind conditions falling within or beyond the no-go zone, highlighting the significance of the fix relative to the cartographic projection.

Below is a brief summary of the main impacts in results after rectifying the VISIR-2 code:
- Reduction in the entity of the percentage savings ($CO_2$ for ferry and time for sailboat)
- Increased number of non-FIFO sailboat routes (from 1 to 5)
- Some route topology changes seen in the sailboat bundles (Fig.13.b and Supplement)
- Improved agreement with openCPN for upwind sailing (errors reduced from about 9 to 3%)

However, the qualitative findings from the manuscript remained unchanged.

These fixes involved revisiting the source code (a list of changes is provided in the following table) and recalculating all affected computations, as well as modifying several figures (Fig. 9, 11-14, A1), tables (Tab.1,5-11), and text accordingly, even in the Supplementary Material. A new section (2.2.1) introduces the need and features of the cartographic projection used. The changes in the source code files and functions are listed in the document with the overview of changes provided along with this review.

| VISIR-2 module | files | functions |
|---|---|---|
| Grafi | proc_edges.py
coast_intersection.py
grid.py
save_graph.py ->
graph_postproc_save.py
prov_edges.py | edge_center_calculation()
get_clear_edges()
coast_intersect()
check_edge()
coast_proximity()
Grid()
graph_save() |
| Campi | edge_Waves.py
edge_Currents.py
edge_Currents_analytic.py
edge_Wind.py | edge_wave_computation()
edge_curr_computation()
edge_wind_computation()
analytic_curents() |
| Utilita | read_namelist.py
ProjectorClass.py
PlotProjectiorClass.py (new file) | namelist_postProc()
ProjectorClass()
PlotProjectiorClass() |
| Tracce | get_trackMetrics.py | trackMetrics() |
| Visualizzazioni | MAIN_Visualizzazioni.py
bundles.py
isolines.py
mapPlot.py

netCDF_generator.py
plot_graph_utils.py
reproduce_gmd_2023_plots_and_t
ables_utils.py | MAIN()
MAIN(), add_track()
isolinesContour()
envFieldPlot(), load_shoreline(),
plot_crt(), plot_wave(),
plot_wind()
makeNetCDF(), make_isolines()
graph_show()
isolinesContour(),
load_shoreline(), plot_crt(),
plot_wave(), plot_wind() |
| Validazioni | analytic_results.py
benchmark_results.py
job_dictionaries.py | show_analytic_results()
show_benchmark_results()
dictionary entry |

Updated figures:

[revised manuscript text omitted]

The conclusion section is too long and should be a synopsis of the contribution and highlights of the results that a reader should and must take away from reading the publication. No new results or new discussion should be present in the conclusion.

Thanks for specifying this.

To address it, we have created a Discussion section between the Results and the Conclusions sections. Furthermore, we have anticipated some remarks to the Methods and Results

sections. We have ensured that the Conclusions section does not include any new information.

**Technical Corrections**

Unless otherwise specified, the subsequent corrections have been applied to the preprint lines mentioned by the Referee.

Ln 6: A least-CO2 algorithm in the presence of
Fixed, thanks.

Ln 12: Two-digit percentage? Two-digit quantity? Suggest clarification on this improvement as its unclear on the units / tangibility of statements. Two-digit pounds of CO2 emission for example is not as impressive as say two-digit percentage of overall CO2 expenditure.
Thanks for noting this imprecision. A more accurate statement, corresponding to Eq.22, would be "a two-digit percentage of overall emissions", and it has been revised both here and throughout the manuscript.

Ln 14: 3% shorter as measured by time, or distance? Based on the authors' prior words "path elongation", it is confusing to the reader to tout a 3% shorter result.
According to the results in Tab.11, it is 3% in terms of duration: so faster and not shorter. Instead, due to diversions from the least-distance route (cf. Fig.14a), path length increases.

Ln 17-18: If you are using winds, then meteorology should be included in the list of knowledge bases pulled from
Added, thanks.

Ln 37: CE-Ship model is an undefined concept or acronym, it also doesn't seem to be used elsewhere so no need to use the acronym unless it is most commonly known by that name
CE-Ship is CE Delft's proprietary GHG emissions model for the global shipping sector. However, we are unable to expand the CE acronym. A concise description of the model was provided in the referenced [Faber_2023] paper.
We have provided a short model description and enclosed its name in quotation marks.

Ln 44: Need a reference for this statement. The reviewer agrees the estimates often are in fact in the 2-5% range but these sources are not mentioned here. Suggest including the reference that assesses the fuel savings (on average) to be <10%. Some open literature is easily searchable /citable for 2-5% estimates.
The presented percentage savings were derived from the referenced papers. Specifically, both the 50% and the 10% figures were sourced from [Bouman_2017, Fig.2]. Regarding the suggested 2-5% range, in the peer-reviewed literature we found a work by [Miola_2011], with values between 0.2 and 3.9%, while, for eastbound routes of a Panamax bulk carrier in the North Atlantic, [Mason_2023a] reported values from 2.2 to 13.2%.
We have added an entire subsection (6.1) devoted to a critical comparison of percentage savings.

Ln 105: risk attitude seems an unusual term, the more common scientific term in the literature on human cognition in the context of decision support systems refers to it as risk propensity

Thank you, we have now substituted it with your suggested alternative.

Ln 141-141: Suggest renaming STW and SOG to be velocity through water and velocity over ground, as it is contradictory to state you are taking the vector sum of speed with something else (in this case ocean current). In a similar vein, the authors state the forward speed F is a vector. Speed is only the magnitude, hence it's recommended such quantities take on the definition /name of velocity, rather than a speed – forward velocity F in this example.

Thanks for pointing out this inconsistency.

Apart from fixing the terms velocity or speed as needed throughout the manuscript, we have now introduced a more uniform vector notation. For example, the vector whose magnitude is the speed over ground (SOG) is now called $\overrightarrow{G}$= SOG $\hat{e}$. The bold font used so far for vector quantities has been replaced by a vector arrow. Fig.1 has been revised to reflect these updates.

Ln 183, shouldn't this be modulo 2*pi radians or 360 degrees?

In the code, the check on the no-go zone is actually performed on the absolute value of the relative angle of heading with respect to wind (Navi/VesselClasses/SailboatClass.py). This quantity is restricted to the range [0, 180] only.

We have revised the mentioned Eq.7 accordingly.

Ln 260 Collinear in what transformation space/projection? Lines of constant bearing (rhumb line) or great circle lines?

Upon the adoption of a cartographic projection, the VISIR-2 graph continues to be generated from an equidistant lat/lon grid, which is subsequently projected onto a Mercator map. Subsequently, edge orientations are computed based on distances in the projection space. Thus, the graph edges are by construction rhumb lines.

A computational aspect regards the fact that vertical spacing in a Mercator projection is uneven and increases with latitude. However, in the VISIR-2 code (*gen_edge.py*), edges are defined as collinear if they share the same ratio of horizontal to vertical grid hops. Hence, the pruned multi-hop edges may represent directions that (slightly) differ from those of the single-hop ones. Consequently, we introduced the term "quasi-collinear" edge to refine our description of the graph.

Pruning such quasi-collinear edges remains beneficial for creating a lighter graph devoid of longer edges, thus resulting in a more accurate representation of environmental fields.

We have updated Fig.2 caption to inform the reader also about the shape of the graph grid and clarified "quasi-collinear edges" in the text of Sect.2.2.3.

[Figure]

**Figure 2.** Graph stencil for $\nu = 4$: a) grid in spherical coordinates with $\Delta x$ resolution along both latitude and longitude; b) Mercator projection, with uneven resolution along $y$ or $x$, with graph edges (black thick and grey dashed lines) and (light blue) angles relative to due North. The $y$ spacing is here shown as constant but, over a large latitudinal range, does vary. In VISIR-2, just the $N_{q1}(\nu)$ dark grey nodes (cf. Tab. 1) are connected to the origin while, in VISIR-1, all $\nu(\nu+1)$ dark or light grey nodes were connected.

Ln 282-285, this provided approach works for Cartesian measurements and coordinate systems, but the proposed research application is that of nautical navigation. How do the authors attend to this? At a minimum, a projection is required somewhere.
Yes, correct. As outlined above, this is now addressed via projecting both graph edges and shoreline elements, followed by conducting a search for intersections within a circle centred on the edge barycenter, in the projection space.
See revised Fig.2 above and changes in the coast_intersection.py file of the source code.

Ln 312, shouldn't a ceiling function be used in the interest of safety of navigation? Drivers and sailors with differing risk propensities may have different agreement with recommendations if they are pessimistic vs optimistic edge weight estimation.
We believe the Reviewer is here referring to the estimation of edge delay. We understand that their proposal is to use systematically biased estimations of this quantity, depending on the user's  risk propensity. However, Sect.2.3.2 refers to the interpolation of environmental fields. They only indirectly and in a nonlinear fashion, through Eq.17, contribute to the edge delay or other edge weights (such as $CO_2$ emissions). Thus, the spatial interpolation scheme would not be reflected in a predictable way into the local sailing speed.
We have now clarified this in the latter part of Sect.2.3.2.

Ln 315 do the authors mean "the same outcome" ? Weather is highly nonlinear though so what analyses has been done to understand the tradeoffs for these two interpolation schemes in a dynamic nonconvex environment?
To test the two interpolation schemes, we have generated fields of varying curvature and edge lengths on different hypersurfaces of the three dimensional space to simulate both various field nonlinearities and graph grid resolution. Regardless of the interpolation option chosen (Sint=0 or Sint=1), the results converge towards the same value as the resolution increases. For specific transects of the hypersurface, either the Sint=0 or Sint=1 scheme yields an outcome closer to the asymptotic value. This suggests that neither scheme demonstrates a consistent superiority over the other in terms of fidelity.
On the other hand, upon closer examination, the computational performance was found to be contrary to what was stated in the preprint. More precisely, due to its application for a

significantly lower number of times (specifically, at each node rather than at each edge), Sint=0 proves to be faster than Sint=1. Consequently, we have established the computationally faster option, Sint=0, as the new default interpolation scheme of VISIR-2. Supplement's Sect.S0. has been introduced to evaluate the impact of the two schemes:

[Figure]

**Fig. S0** a)-c) Test hypersurfaces (shaded in grey) and graph edges (coloured lines and markers indicating the edge head). d)-f) Edge representative values of the different hypersurfaces for both Sint=0 (circles) and Sint=1 (triangles).

Furthermore, Sect. 2.3.2. and Fig.5 have been updated:

[Figure]

**Figure 5.** Spatial interpolation in VISIR-2. a) The squares represent grid nodes of the environmental field $\varphi(x)$, and the filled circles graph grid nodes. A graph edge is depicted as a magenta segment. b) Transect of a) along the edge direction $\hat{e}$, with the interpolator of $\varphi$ as a grey solid line. The (0,1) subscripts refer to the value of the Sint parameter while $h$ and $t$ to the edge head and tail, respectively.

In the model source code, Sint=0 is now set as the default interpolation scheme.

Ln 326 The sentence ordering makes it seem that VISIR-1 is the improvement of Dijkstra for dynamic edge weights when I believe the authors intend to credit Orda & Rom 1990.
Thank you for pointing that out. Indeed, our statement was unclear and partially incorrect. A more accurate one would be as follows:
Dijkstra's original algorithm of 1959 exclusively accounted for static edge weights. When dynamic edge weights are present, [OrdaRom_1990] demonstrated that, in general, there are no computationally efficient algorithms. However, they also showed that, upon incorporating a waiting time at the source node, it is possible to keep the algorithmic complexity of a static problem. If the rate of variation of the edge delay is never smaller than -1, waiting is not even needed. This situation, referred to as "FIFO" (First In, First Out), has

been utilised for coding a dynamic Dijkstra's algorithm since VISIR-1 and continues to be implemented in VISIR-2.
We have made revisions to the beginning of Sect. 2.4.1 to ensure it aligns more effectively with the enhanced explanation.

Ln 341,347 FIFO-hypothesis is the correct English spelling.
Fixed, thanks.

Ln 416 "straight" by what measurement? Constant bearing/dead reckoning, or shortest path on a sphere?
Ultimately, route legs correspond to graph edges. Building on the Referee's previous point regarding cartographic projection, we now calculate the orientation of these edges not in spherical coordinates but on a Mercator map. Hence, in this context, straight navigation will refer to segments with a constant bearing between the locations of the edge nodes.
A sentence to make this clear to be added at the end of Sect.2.6.

Ln 665-666: From layman's understanding, your findings confirm those of prior work in bibliographic citation [Sidoti et al., 2023] in importance considering both current and leeway for sailboat routing optimization. Can you be more specific regarding what "this" refers to when the authors state "This is, ..., the first of its kind assessment"?
Your observation is accurate: [Sidoti_2023] already considered both currents and leeway in sailboat routing. Although employing a distinct methodology, Sidoti's one precedes VISIR-2.
We have revised our text to avoid attributing such precedence to VISIR-2, which instead belongs to the work by [Sidoti_2023], which is now acknowledged also in the Conclusions.

---

## Author Response (AR2)

**VISIR-2: ship weather routing in Python**

Mannarini, Salinas, Carelli, Petacco, Orović

submission v04 - April 9th, 2024

Dear Editors,

I am submitting the revised production files for our manuscript.

Below are the changes made compared to the previous version (v03 submitted on February 27th, 2024):

- Abstract: fixed two sentences for avoiding the repetition of word "distribution" and for swapping the shorthand and more technical definition of $CO_2$ saving
- Sect.2.1: Using definition Eq.2 into Eq.7
- Fig. 2: replaced \psi_s with \psi_h as requested in Reviewer 1's minor comment 2. Note that this fix was inadvertently omitted in v03
- Fig.3: avoided the red-green line colour issue by using gold and red
- Fig.10: avoided the red-green line colour issue by adding a linestyle
- Fig.11.a: avoided the red-green line colour issue by adding a linestyle
- Fig.11.b: avoided the rainbow colormap issue by using a plasma colormap
- Fig.13.a: avoided the red-green line colour issue by adding a linestyle
- Fig.13.b: avoided the rainbow palette issue by using a plasma colormap
- Sect.5.2.2: Fixed a potentially misleading expression relative to the savings in upwind sailing
- Sect.5.2.2: split paragraph relative to failed routes from paragraph relative to time savings
- Conclusions: avoided rounding of percentage savings for sailboat, using values consistent with those in abstract

Therefore, in addition to the modifications requested by the Topic editor, there are a few minor changes aimed at enhancing clarity in the text, which I hope will be acceptable. I am also attaching a diff file to document these amendments.

Kind regards,
Gianandrea Mannarini

[revised manuscript text omitted]